

**A robust clustering algorithm for analysis of composition-dependent**
**organic aerosol thermal desorption measurements**
Ziyue Li[1], Emma L. D'Ambro[2,3,a], Siegfried Schobesberger[2,4], Cassandra J. Gaston[2,b], Felipe D.
Lopez-Hilfiker[2,c], Jiumeng Liu[5,d], John E. Shilling[5], Joel A. Thornton[2,3], Christopher D. Cappa[1,6]
[1] Atmospheric Science Graduate Group, University of California, Davis, CA, USA
[2] Department of Atmospheric Sciences, University of Washington, Seattle WA, USA
[3] Department of Chemistry, University of Washington, Seattle WA, USA
[4] Department of Applied Physics, University of Eastern Finland, Kuopio, Finland
[5] Atmospheric Sciences and Global Change Division, Pacific Northwest National Laboratory,
Richland WA, USA
[6] Department of Civil and Environmental Engineering, University of California, Davis, CA, USA
[a] Oak Ridge Institute for Science and Education, US Environmental Protection Agency, Research
Triangle Park, NC, USA
[b] Rosenstiel School of Marine & Atmospheric Science, University of Miami FL, USA
[c] TofWerk AG, Thun, Switzerland
[d] Now at: School of Environment, Harbin Institute of Technology, Harbin, Heilongjiang, China
## Abstract
One of the challenges of understanding atmospheric organic aerosol (OA) stems from its complex
composition. Mass spectrometry is commonly used to characterize the compositional variability
of OA. Clustering of a mass spectral data set helps identify components that exhibit similar
behavior or have similar properties, facilitating understanding of sources and processes that
govern compositional variability. Here, we developed a novel clustering algorithm, Noise-Sorted
Scanning Clustering (NSSC), and apply it to thermal desorption measurements from the Filter
Inlet for Gases and AEROsols coupled to a chemical ionization mass spectrometer
(FIGAERO-CIMS). NSSC provides a robust, reproducible analysis of the FIGAERO temperature-
dependent mass spectral data. The NSSC allows for determination of thermal profiles for
compositionally distinct clusters, increasing the accessibility and enhancing the interpretation of
FIGAERO data. Applications of NSSC to several laboratory biogenic secondary organic aerosol
(BSOA) systems demonstrate the ability of NSSC to distinguish different types of thermal
behaviors for the components comprising the particles along with the relative mass contributions
and chemical properties (e.g. average molecular formula) of each cluster. For each of the systems
examined, more than 80% of the total mass is clustered into 9-13 clusters. Comparison of the
average thermograms of the clusters between systems indicate some commonalty in terms of
the thermal properties of different BSOA, although with some system-specific behavior.
Application of NSSC to sets of experiments in which one experimental parameter, such as the
concentration of NO, is varied demonstrates the potential for clustering to elucidate the chemical
factors that drive changes in the thermal properties of OA. Further quantitative interpretation of
the clustered thermograms followed by clustering will allow for more comprehensive
understanding of the thermochemical properties of OA.



## 1. Introduction

Atmospheric particles are composed of hundreds to thousands of individual compounds
(e.g., Hamilton et al., 2004; Goldstein and Galbally, 2007), reflecting the many different sources
and the variety of chemical pathways that lead to their formation and growth. Various mass
spectrometry (MS) methods provide for characterization of this compositional variability, among
other techniques. Individual MS methods yield different insights into particle composition,
dependent upon the chemical selectivity of the method. Application of various data reduction
methods, such as clustering or matrix factorization, helps to reduce the inherent compositional
complexity and develop understanding of the sources and chemical transformations that
determine particle composition. Clustering and matrix factorization are complementary methods.
In this work, we develop and apply a new clustering method to measurements of the evolved gas
composition derived from thermal desorption of organic aerosol, specifically to measurements
from the Filter Inlet for Gases and AEROsols (Lopez-Hilfiker et al., 2014) coupled with chemical
ionization mass spectrometry (Lee et al., 2014) (FIGAERO-CIMS). The clustering method
developed here facilitates interpretation of variability in organic aerosol composition and
volatility, and how these depend on formation conditions.
Clustering methods applied across many research fields have aided in the interpretation
and understanding of large data sets. Clustering methods work by classifying data into several
groups according to the similarity between one or more properties. In the field of atmospheric
chemistry, clustering methods have been applied to a variety of data types. Examples include:
back trajectories of trace gases (Cape et al., 2000) or particles (Abdalmogith and Harrison, 2005;
Pinero-Garcia et al., 2015), helping to elucidate the origin and transport of pollutants; particle
size distributions, providing information on aerosol emission and formation (Beddows et al., 2009;
Wegner et al., 2012); and, the morphology of and organic functional groups comprising individual
particles, allowing for classification of the types of organic carbon (Takahama et al., 2007).
Beyond the above examples, clustering methods have been extensively applied to the
interpretation of single particle mass spectra, serving to characterize variability in their chemical
composition and identify the sources and extent of chemical processing (e.g., Gaston et al., 2013;
Lee et al., 2015). While clustering is a general method, a variety of specific algorithms have been



developed for application to a given particle mass spectral dataset. The algorithms applied to
analysis of single particle mass spectra include: *K*-means (Giorio et al., 2012; Liu et al., 2013; Lee
et al., 2015); fuzzy *c*-means (Kirchner et al., 2003; Roth et al., 2016); density-based special
clustering of applications with noise (DBSCAN) (Zhou et al., 2006); neural network-based
methods, such as an algorithm derived from Adaptive Resonance Theory (ART-2a) (Song et al.,
1999; Zhao et al., 2008; Giorio et al., 2012); hierarchical clustering (Murphy et al., 2003; Rebotier
and Prather, 2007); and, some combined algorithms (Zhao et al., 2008; Reitz et al., 2016). Each
clustering algorithm has strengths and weaknesses. In some cases, different algorithms are
equally effective and lead to similar categorization of the same data set, while in other cases
quite different results are obtained (Zhao et al., 2008). For example, *K*-means and ART-2a gave
broadly similar results on a regional particle data set (Giorio et al., 2012), and *K*-means performed
as well as a variant of hierarchical clustering method on four particle data sets (Rebotier and
Prather, 2007).
Here, we describe and apply a new clustering method for analysis of combined thermal
desorption-mass spectral measurements of organic particle composition, specifically applied to
data from the FIGAERO-CIMS. FIGAERO-CIMS has been increasingly used in field (e.g. Gaston et
al., 2016; Lee et al., 2016; Lopez-Hilfiker et al., 2016; Mohr et al., 2017; Huang et al., 2018; Le
Breton et al., 2019) and laboratory studies (e.g.Lopez-Hilfiker et al., 2015; D'Ambro et al., 2017;
Wang and Ruiz, 2018) to develop understanding of the molecular composition of organic aerosols.
A key feature of FIGAERO-CIMS is the ability to characterize the thermal behavior of organic
compounds in particles on a near molecular level (Lopez-Hilfiker et al., 2014). The use of chemical
ionization, a relatively soft ionization method, facilitates detection and characterization of both
monomeric and oligomeric parent compounds in organic aerosols. In FIGAERO-CIMS, particles
are collected and then thermally desorbed, with mass spectra of the evolved gases measured as
a function of temperature. This can also be displayed as a thermogram: the concentration of an
ion or sum of ions as a function of desorption temperature. The temperature at which a
thermogram reaches maximum signal, or $T_{max}$, provide information on the volatility, while
particularly broad desorption shapes can indicate thermal decomposition, suggesting the
presence of lower volatility, possibly oligomeric, material (Lopez-Hilfiker et al., 2014). A typical



FIGAERO-CIMS mass spectrum of either ambient or laboratory-generated organic aerosol
consists of hundreds of individual ions and thermograms, (D'Ambro et al., 2018; Lee et al., 2018).
Previous studies using FIGAERO-CIMS provided insights into particle composition, including
the presence of lower volatility material, based on analysis of the thermograms of several major
ions (Lopez-Hilfiker et al., 2014; D'Ambro et al., 2017; D'Ambro et al., 2018; Lee et al., 2018). We
expand on this previous work through the application of cluster analysis to FIGAERO-CIMS
thermograms. Clustering of FIGAERO-CIMS data provides a means to expand the understanding
developed from single-ion thermograms and establish the contributions of different types of
thermograms to the bulk particles. One previous study clustered FIGAERO-CIMS data using the
K-means algorithm using two parameters: the ion molecular weight and the maximum
desorption temperature (Faxon et al., 2018). What distinguishes our work is that we cluster the
thermogram across the entire desorption period for each ion, with ions grouped according to the
similarity of their overall volatility distribution. We have considered the performance of various
clustering algorithms (including K-means), ultimately concluding that a novel variant of the
DBSCAN algorithm developed here, named noise-sorted scanning clustering (NSSC), provides
robust performance and has several advantages over other existing algorithms for FIGAERO-CIMS
data. The NSSC algorithm is applied to several laboratory data sets of secondary organic aerosol
(SOA) formed from various precursors and under various conditions, some are previously
described (D'Ambro et al., 2018). In this work we do not aim to provide comprehensive
interpretation of the resulting clustered thermograms in terms of their thermo-chemical
properties (Schobesberger et al., 2018), only to illustrate the potential of clustering to enhance
interpretation of FIGAERO-CIMS and other similar data.

## 2. Clustering Method Description

Application of a given clustering algorithm to a particular data type involves a number of
steps. Below, we discuss the specific steps for clustering of FIGAERO-CIMS data, including a
description of our noise-sorted scanning clustering algorithm. A brief discussion of other
algorithms is also provided.





**2.1.    Data Preprocessing**
**2.1.1.  Exclusion of anomalous thermograms**
The quality of the data set should be examined prior to clustering. A typical thermogram
exhibits a continuous evolution to a peak, peaking during a temperature ramping period, after
which there is a steady decrease in signal-to-background over time during a constant-
temperature soaking period; the background-corrected signal at all temperatures remains above
zero or around zero within the uncertainties. See section 3.1 for further details of the FIGAERO-
CIMS. An anomalous thermogram, however, contains negative signal with large absolute
magnitude.
Anomalous thermograms should be excluded from the clustering to assure the quality of
the results, although most such thermograms do not end up clustered with other ions.
Anomalous thermograms are identified as follows. (i) Estimate a reference noise level ($\sigma_{ref}$) for
each thermogram as the standard deviation of the last 100 points (corresponding to 500 seconds)
of the thermogram at the end of the constant-temperature soaking period, during which the
signals are usually relatively constant. (ii) Find the minimum in the thermogram and calculate the
average of this and the 50 points (corresponding to 250 seconds) before and after the minimum,
$A_{min}$. (iii) Identify thermograms for which $A_{min} < -3*|\sigma_{ref}|$ as anomalous and exclude these
associated ions from further analysis. In other words, when a thermogram has a valley with
averaged negative values exceeding the magnitude of three times of the reference noise level,
then it is considered anomalous.
Ideally, when anomalous ions are identified the original data would be inspected to identify
the likely origin of the anomalous behavior. Possible origins include problems with background
subtraction when the blank has substantially higher signal levels than the particle samples, which
can happen when there is residual contamination or incomplete separation of ions having the
same nominal mass. It is also possible that the components detected for the same ion are
different for the particle and blank measurements. In the example systems considered here, we
identified up to five anomalous ions out of what is typically a few hundred total ions.



In some cases, it is desirable to compare thermograms between related experiments, for
example the experiments discussed here that investigated the influence of NO concentration on
SOA formation (Section 4.3) and the impact of isothermal dilution on SOA composition and
volatility (Section 4.4). In such cases, ions identified as anomalous for one experiment are
excluded from analysis for all related experiments to ensure consistency.

### 2.1.2. Euclidean Distance

Any clustering algorithm requires a metric to determine the similarity between two
members in the data set. Here, we use the commonly used Euclidean Distance (ED) as the metric.
A smaller *ED* indicates greater similarity. A FIGAERO thermogram has $n$ points, with all
thermograms having an equal number of points in a data set. A data set here is defined as the
collection of thermograms for all individual ions measured for a single desorption event. The *ED*
between two thermograms *a* and *b* is calculated as:

$$ED_{a,b} = \sum_n \sqrt{(a_n - b_n)^2} \qquad\qquad (1)$$

An individual *ED* value is obtained for every pair of ions in the mass spectrum, resulting in an $n$ x
$n$ matrix of *ED* values with the diagonal elements all zero. The signal levels between individual
ions differ substantially, reflecting their relative abundances. Therefore, the *ED* calculation uses
normalized thermograms, allowing for comparison between thermogram profiles irrespective of
signal magnitude. Normalization is achieved by dividing each point of the original thermogram
by the thermogram maximum after smoothing using a 35-point moving average. Use of the
smoothed maximum instead of the unsmoothed maximum reduces the influence of noise on
normalization. In the FIGAERO datasets used in this study, a typical thermogram has a
temperature resolution of $\Delta T \sim 0.7$ °C during the ramping period, and a 35-point smooth
corresponds to smoothing over ~24.5 °C. Typical FIGAERO thermograms exhibit peaks ca. 40 °C
wide, and thus a 35-point smoothing retains the main peak shape while reducing the influence
of noise. In the constant temperature part of the thermogram (soaking period), signal levels
change slowly with time, on average less than 5 % for a 35 points (~3 minutes) period, so a



35-point smoothing is also appropriate. We note that the unsmoothed profiles are those that are
normalized; smoothing relates only to determining the maximum signal values used for
normalization.
The *ED* calculation from Eqn. 1 gives equal weight to all points in the thermogram. However,
in a FIGAERO thermogram, equal weighting may not be appropriate. The desorption process has
two stages, ramping and soaking, with the soaking period comprising approximately 70% of the
time points in thermograms. However, most thermograms are featureless in the soaking period.
In contrast, many thermograms exhibit a peak, or some otherwise characteristic behavior, in the
ramping period. Since the behavior in the ramping period provides greater information as to the
overall similarity between individual thermograms, we recommend down-weighting the soaking
period such that the ramping and soaking periods ultimately carry approximately 4:1 weight in
the calculation of the *ED*. We do not recommend completely excluding the soaking period as this
period still carries informational content (Schobesberger et al., 2018). Specifically, in calculating
*ED* we use all data from the ramping period while down-weighting the data in the soaking period
by calculating and using ten-point averages.
In summary, we calculate the *ED* based on the following steps: (i) smooth the original
thermogram (with absolute signal) to find the maximum value; (ii) normalize the original
thermogram to the smoothed maximum; (iii) average every 10 points in the soaking period; and
(iv) calculate the *ED* between every two normalized, down-weighted thermograms.

### 2.1.3. Dealing with noise

Noise is an inherent property of any measurement. Noise in the FIGAERO thermograms
results from various sources, including detector noise, background subtraction, and imperfect
fitting of mass spectra. Noise influences the ED calculated between two thermograms, typically
increasing the *ED*. Here, the level of noise, $\xi$, is characterized for each thermogram by calculating
the average difference between the smoothed and unsmoothed normalized thermograms for
the ramping period. The use of only the ramping period in assessing the noise level is consistent
with the generally more characteristic behavior compared to the soaking period. The use of the
normalized thermograms, rather than absolute, allows for comparison of noise between
thermograms.



The noise level generally varies inversely with the fractional mass contribution of the ions,
illustrated for a case study of the $\alpha$-pinene + OH SOA (Experiment 1 in **Table 1** and **Figure 1**). This
indicates that ions contributing more to the total signal generally have a lower noise level.
Detector noise is nominally independent of ion identity, and thus the low-signal ions have
enhanced $\xi$ after normalization.
Discussed further in section 2.3, clustering algorithms often perform poorly when overly
noisy data are included in the clustering. This is especially the case for algorithms such as k-means
and partitioning around medoids, which assign all the members to a cluster. The inclusion of
overly noisy peaks might obscure the underlying structure of clustered thermograms. Noisy
thermograms are identified as follows. First, the 5% of ions having the lowest noise are identified.
The $\xi$ value of the noisiest ion from this subset of low-noise ions is defined as the reference noise
level, $\xi_{ref}$. Small differences in the choice of this threshold (e.g. using the lowest 7% of ions) do
not materially influence the results. Ions for which $\xi_n > 3 \cdot \xi_{ref}$ are considered noisy and excluded
from the initial clustering. For the experiments we examined, there are 88-120 out of ~300 ions
left after noise screening, contributing 83.5% - 92.5% to the total particle mass.
**2.2.    Noise-sorted Scanning Clustering (NSSC)**
**2.2.1.  Algorithm description**
The noise-sorted scanning clustering (NSSC) algorithm developed here is a variant of the
commonly used DBSCAN. In NSSC, identification and clustering of thermograms occurs based on
their similarity to seed thermograms. When the *ED* between a given thermogram and the seed is
less than a specified *ED* criterion ($\varepsilon$) the two members belong to the same cluster. Importantly,
in NSSC the selection of the seed thermograms occurs based on their respective noise levels. The
least noisy thermogram is selected as the initial seed, the next noisiest is selected as the second
seed (assuming it is not already clustered), and so on. We have found that low-noise
thermograms typically have more well-defined and characteristic shapes and comprise a
substantial fraction of the total mass. The choice to select seeds based on the noise level leads
to overall more robust and reproducible clustering compared to random selection of seeds.



The optimal value of the distance criterion, $\varepsilon$, is not known *a priori*, but must be determined
by the user, discussed in Section 2.2.3. A valid cluster must contain at least $N_{min}$ members,
inclusive of the seed. We use $N_{min}$ = 2. Consideration and inspection of individual unclustered
thermograms exhibiting unique behavior occurs as a post-clustering process (Section 2.2.2).
The flow of the noise-sorted scanning clustering algorithm is shown in **Figure 2**, and
summarized here. Clustering proceeds in two rounds. For the initial round, the thermograms are
sorted by the noise ($\xi$), and the ED values between all pairs of thermograms are calculated
accordingly. All of the thermograms are identified according to whether they have been already
used as seeds (SEED = 0 or 1, with 1 for thermograms used as seeds) and whether they have been
already included in a cluster (CLUSTER = 0 or 1, with 1 for already clustered thermograms). At the
start, SEED = 0 and CLUSTER = 0 for all thermograms. Clustering begins using the least noisy
thermogram having SEED = 0 and CLUSTER = 0 as the initial seed. The state of that seed is then
changed to SEED = 1. All thermograms having $ED < \varepsilon$ for that seed and with CLUSTER = 0 are
identified from the *ED* matrix; these thermograms are considered neighbors of the seed
thermogram. If the number of neighbors plus the seed is greater than or equals $N_{min}$, the cluster
is valid and stored, with the states of all the thermograms in the cluster changed to CLUSTER = 1.
Otherwise, the cluster is dismissed, and CLUSTER = 0 for all the members. In this case, the current
seed (with SEED = 1 and CLUSTER = 0) will no longer be used as a seed in the future steps but can
still end up clustered as a neighbor in the other clusters. The above steps are repeated until all
the thermograms have either SEED = 1 or CLUSTER = 1.
Because a cluster must have at least $N_{min}$ elements, not all the thermograms may end up
clustered. Some of these unclustered thermograms may nonetheless have very similar shapes to
the clustered thermograms. Here, an iterative, second round of clustering potentially adds these
initially unclustered thermograms to the initial clusters, using the signal-weighted average
thermograms for the clusters from the first round as the initial seeds. A matrix of *ED* values is
calculated between the individual unclustered thermograms and the new seeds. For each
unclustered thermogram, the minimum *ED*, corresponding to only one of the seeds, is identified.
When this minimum *ED* is less than $\varepsilon$, the unclustered thermogram is added into that cluster. A
new signal-weighted average thermogram for the cluster is calculated and this process repeats





until no additional unclustered thermograms can be added to existing clusters. The mass
contribution of the remaining unique unclustered thermograms after this second round can be
substantial or negligible, ranging from <0.05% to 2.6% in the experiments presented here, and
depends largely on the choice of ε. Some of these unclustered thermograms are defined as
additional one-member clusters, discussed in the following section.
**2.2.2.  Post-clustering Processes**
After thermograms are clustered, we perform two post-clustering analyses to better
understand the whole data set: 1) identifying additional one-member clusters and 2) sorting of
the clusters.
Some of the remaining unclustered thermograms have significant individual mass
contributions and should be considered as one-member clusters. The criterion of "significant"
mass contribution is user-defined. We recommend determining the significance criterion as
follows: (i) sorting all the ions (before the noise-filtering process) from largest to smallest
individual mass concentration; (ii) calculating the cumulative mass fraction for this sorted list;
and (iii) defining as "significant" all those ions contributing to a cumulative mass contribution up
to 80%.
The number of significant ions in a data set depends on the specific chemical system,
varying from only a few to tens of ions. Significant unclustered ions are identified as additional
one-member clusters. In some cases, the thermograms for these one-member clusters are
unique compared to the previously identified clusters. In others, their shapes are visually similar
to the previously identified clusters but where the one-member clusters are sufficiently distinct
that they were not clustered. For the purpose of automation, these one-member clusters are all
included in the final clustering results and the number of one-member clusters serves as one of
the parameters to determine the optimal ε. User can also choose to exclude them or some of
them manually from the final clustering results based on their judgement. For the example
systems considered in Section 4, there are only a few one-member clusters (ranging from 0 to 4),
if any, for the optimal ε used.



Sorting of clustered thermograms facilitates visual presentation and identification of the
similarities and dissimilarities among the clusters. The specific method of sorting can be varied
depending on the application and system under consideration. Here, we use the temperature
where 50% of the mass is desorbed ($T_{m50}$) for the weighted-average cluster thermogram as a first
criterion. The $T_{m50}$ is typically similar to, but slightly larger than the temperature at which the
signal reaches a maximum. As such, the $T_{m50}$ is approximately related to the saturation vapor
pressure of the desorbing compound, at least for compounds that desorb directly (e.g., Lopez-
Hilfiker et al., 2014). When two or more clustered average thermograms have identical $T_{m50}$, a
rare but occasional occurrence, they are further sorted by $T_{m75}$, the temperature where 75% of
the mass is desorbed. The temperature difference between $T_{m50}$ and $T_{m75}$ indicates the slope of
the thermogram between these two temperatures, with larger values indicating slower decay.
Therefore, these two parameters generally illustrate the shape of a thermogram. The $T_{m50}$ and
$T_{m75}$ are determined by calculating the cumulative desorbed mass and finding the temperatures
where 50% and 75% are reached.
The sorting process tends to organize the cluster-specific thermograms such that clusters
having lower peak temperatures (lower $T_{m50}$) and steeper downslopes after the peak (lower $T_{m75}$)
come first. Thermograms of this type are indicative of major contributions from higher-volatility
monomers (Schobesberger et al., 2018). Thermograms having higher $T_{m50}$ generally have broader
peaks, and shallower downslopes, indicative of substantial contributions from low-volatility
compounds or decomposition of oligomers. Further discussion of the interpretation of
thermogram shapes is provided in Section 3.2.
### 2.2.3.  Choosing the optimal ε
NSSC is a distance-based clustering method, so the choice of the distance criterion, ε, is a
crucial step. For small ε, members within a cluster have high similarity, but few thermograms end
up clustered. In contrast, for large ε the majority of the thermograms are clustered into only a
few clusters having comparably low intra-cluster similarity. The choice of the optimal ε value is
guided here by consideration of several parameters that vary with ε. The overall aim is to
simultaneously (i) minimize the unclustered mass fraction ($f_{m,unclustered}$) while (ii) maximizing the



number of clusters ($N_c$) having two or more members and (iii) minimizing the number of one-
member clusters ($N_{c,one}$) yet (iv) maintain inter-cluster separation ($R_{interClst}$).

In general, $N_c$ increases with $\varepsilon$ for small $\varepsilon$ because more thermograms of different shapes

get clustered and fewer thermograms remain unclustered. As $\varepsilon$ further increases, some clusters
are combined and a greater number of thermograms are assigned to a single cluster.
Consequently, as $\varepsilon$ increases the $N_c$ generally increases, reaches a maximum level, and then
decreases. The maximum $N_c$ and the $\varepsilon$ at which the maximum occurs depends on the exact size
and the properties of dataset being examined. We have found that a typical SOA system usually
has 9-13 distinct thermogram clusters. We recommend selecting an $\varepsilon$ that provides for $N_c$ at or
near the maximum as this captures the greatest number of thermogram types.

The mass fraction of unclustered thermograms, $f_{m,unclustered}$, includes only the unclustered

thermograms that were not excluded based on the noise filtering. In general, a smaller $f_{m,unclustered}$
is preferable as this indicates a greater amount of the OA mass is included in a cluster (including
one-member clusters). The $f_{m,unclustered}$ generally decreases with $\varepsilon$, then plateaus above a certain
value of $\varepsilon$; ideally this plateau occurs at $f_{m,unclustered} = 0$. The $\varepsilon$ where the plateau starts is indicated
as $\varepsilon_{MF}$, where MF stands for mass fraction. Given that significant one-member clusters are
allowed, the unclustered thermograms that remain above $\varepsilon_{MF}$ have individually small mass
contributions and are either truly unique in their shapes or have a sufficiently high noise level
that they cannot be clustered, even after the noise-screening process. We generally recommend
selecting $\varepsilon \geq \varepsilon_{MF}$ to minimize the unclustered mass.

The number of one-member clusters, $N_{c,one}$, generally decreases with $\varepsilon$, as these ions are

incorporated into multi-member clusters. Ideally, these one-member clusters would exhibit clear,
visually distinct behavior compared to other one-member clusters and to multi-member clusters.
However, we find this is often not the case, especially at smaller $\varepsilon$. Thus, the number of one-
member clusters should generally be minimized; we suggest $N_{c,one}$ be held to five or fewer in
general.

The inter-cluster separation parameter, $R_{interClst}$, characterizes the dissimilarity between

clusters, and is the ratio between the average inter-cluster distance ($ED_{seed,avg}$) and $\varepsilon$, where:


$$R_{interClst} = \frac{ED_{seed,avg}}{\varepsilon} = \frac{\sum_{i=1}^{N_{c,total}} \sum_{j=1}^{N_{c,total}} ED_{seed,i,j}}{N_{c,total} \cdot (N_{c,total}-1) \cdot \varepsilon}$$ (2)


and $ED_{seed,i,j}$ is the distance between the seeds for the different clusters $i$ and $j$ and $N_{c,total} = N_c +$
$N_{c,one}$. For a 2D data set, the seed can be visualized as the center of a circle and $\varepsilon$ the radius of
the circle. Thus, when $ED_{seed,i,j}/\varepsilon < 2$, the two circles defining the boundaries of these two clusters
have overlapping areas. Good separation (i.e. cluster dissimilarity) is indicated when $ED_{seed,i,j}/\varepsilon >$
2. Although our data set is more than two dimensions, this illustrates the idea of establishing the
level of similarity (or dissimilarity) between clusters, i.e., the extent to which they are unique. We
recommend selecting an $\varepsilon$ that results in $R_{interClst} \geq 2$, when possible.
All four parameters should be considered when determining the optimal $\varepsilon$. Consideration
of the parameters individually may not result in the same optimal $\varepsilon$. Ultimately, the user must
consider each parameter and aim to select an optimal $\varepsilon$ that balances the different information
provided in each parameter. This can be achieved by plotting the above parameters as a function
of $\varepsilon$, and then selecting as the optimal value the $\varepsilon$ that results in (i) a small $f_{m,unclustered}$ with (ii) $N_c$
near the maximum and (iii) a small $N_{c,one}$ and (iv) $R_{interClst}$ near or above two. In addition, visual
comparison of the clustering results, illustrated as the average thermogram of each cluster, can
be helpful. For the example data considered below, we find that the optimal $\varepsilon$ tends to fall within
a relatively narrow range of values.
**2.3.    Alternative Clustering Methods**
We have alternatively considered the performance of some of the most commonly used
clustering algorithms (k-means, k-medoids, mean-shift, DBSCAN) and a less-commonly used one
(FPClustering (Gonzalez, 1985)) for interpreting FIGAERO-CIMS observations. The clustering
methods considered are summarized in **Table 2**, with some of their pros and cons listed, and
described in further detail in Appendix A. We discuss them briefly here in the context of FIGAERO-
CIMS data. All the methods considered require input of at least one key user-specified parameter.
These parameters and the associated clustering algorithms can be generally classified into two
categories: number-based and distance-based. Number-based clustering algorithms require
specifying the desired number of retrieved clusters; this includes k-means and k-medoids.



Number-based algorithms usually assign all members to clusters. The extent of similarity among members of a cluster can vary greatly since there is no strict distance criterion for each cluster. When applied to FIGAERO-CIMS thermograms, we have found these number-based algorithms are particularly sensitive to the presence of noisy members and the initialization method. In contrast, some clustering algorithms require specification of distance (similarity) criterion. This includes the mean-shift, DBSCAN, and our NSSC algorithms. These distance-based algorithms need not cluster all members of the initial population and generally emphasize intra-cluster similarity or the density of the points. The methods differ in terms of the method used for selection of the initial seed or center and the extent to which they emphasize point density versus cluster similarity. Noisy members tend to naturally be excluded from any clusters.

Most of these clustering algorithms, including k-means, k-medoids, and mean-shift, are initialized with a random choice of the initial cluster centers (or seeds). For large data sets, this randomness usually leads to different results of clustering with different runs. The extent to which this impacts analysis and clustering of FIGAERO-CIMS data is considered using SOA from the $\alpha$-pinene + OH SOA system as the case study (Section 4.1). For the FIGAERO-CIMS data we find that the various clustering results exhibit a moderate sensitivity to how the initial seeds are selected for all of these algorithms, although the final clusters are generally similar between different runs for the same input parameter. This may reflect either the relatively small size of the data set (~300 members originally and ~100 members after noise screening) or that there are generally characteristic peak shapes with overall good separation. However, some differences between independent clustering runs result, which is undesirable. For FIGAERO-CIMS data we know that not all thermograms are of equal quality, i.e. they have different noise levels reflecting in part their different overall contributions to the total mass. The standard clustering methods do not account for this information. The NSSC algorithm developed here takes into account this measure of data quality and uses it to identify the seeds for clustering. This provides for an entirely reproducible clustering and generally emphasizes the behavior of the ions that contribute most to the FIGAERO-CIMS signal while still allowing for consideration of contributions of low-signal ions.



We find that different clustering algorithms can result in similar numbers of clusters with
the cluster-averaged thermograms having visually similar shapes when each is run with
appropriate user-selected parameters, although the details and robustness of each cluster vary
method by method. The "appropriate" parameters however are different from the "optimal"
parameters. There is usually different guidance for different algorithms on how to find the
optimal parameters that result in the greatest similarity within clusters and dissimilarity among
clusters. In the case of k-medoids, for example, the average silhouette indicates an optimal
number of clusters of two for the case study system. Yet, this is certainly too few clusters based
on the other methods.
In summary, we propose NSSC as the preferred algorithm in dealing with the FIGAERO data
set based on: (i) the ability to generate similar results as the other commonly used clustering
algorithms; (ii) good reproducibility and stability of results due to accounting for the noise of
individual thermograms; (iii) good control over the similarity within the clusters by using a
user-definable distance criterion; and (iv) a capability to identify unique thermograms as
one-member clusters.

## 3. FIGAERO Measurements and Experiments

### 3.1.    Instrument and experiment description

The FIGAERO-CIMS instrument has been described previously in detail (Lee et al., 2014;
Lopez-Hilfiker et al., 2014). A brief description is provided here, with some additional details in
the Supplemental Material. The FIGAERO-CIMS measures the evolved gases from filter-collected
particles during temperature programmed thermal desorption. Thermal desorption of particles
occurs in two-stages: a "ramping" and "soaking" period. During ramping, the temperature
increases from room temperature to 200 °C, typically at 10 °C min$^{-1}$. Most OA mass desorbs
during the ramping stage. The temperature is held at 200 °C for ca. 30–40 mins during the soaking
period to facilitate evaporation of the remaining, low-volatility organic mass from the filter. The
evolved gas-phase compounds are measured using CIMS with the iodide (I$^-$) reagent ion,
appropriate for characterization of generally highly oxygenated components comprising most
secondary organic aerosol (Lopez-Hilfiker et al., 2016; Isaacman-VanWertz et al., 2017; Lee et al.,





2018). The resulting signal or mass concentration versus temperature (or equivalently time)
curves for each ion constitute a thermogram. All individual thermograms are background
corrected by subtracting the observed thermograms from appropriate blank experiments. The
overall bulk thermogram is obtained by summing together the individual thermograms.

Several example applications of the clustering on FIGAERO-CIMS data are discussed in

Section 4. These cover laboratory experiments on SOA derived from: (1) OH + $\alpha$-pinene and (2)
OH + $\Delta-$3-carene, both at low-NO$_x$ conditions; (3) OH + $\alpha$-pinene as a function of [NO]; and (4)
O$_3$ + $\alpha$-pinene, but where the SOA is allowed to isothermally evaporate at 80% RH for varying
amounts of time prior to thermal desorption. These experiments are summarized in **Table 1**, with
further details in the Supplemental Material and associated publications (D'Ambro et al., 2018;
D'Ambro et al., 2019); all data are publicly available (Cappa et al., 2019). All the experiments were
done in a 10.6 m$^3$ Teflon environmental chamber at Pacific Northwest National Laboratory (PNNL)
(Liu et al., 2012; Liu et al., 2016).
**3.2.   General interpretation of FIGAERO-CIMS thermograms**

This work focuses on development of the clustering method, rather than on interpretation

of the FIGAERO-CIMS thermograms; an illustrative thermogram is shown in **Figure 3**b. However,
discussion of the clustering results is aided by a general understanding of how FIGAERO-CIMS
thermograms have been previously interpreted. Ions contributed by semi- and low-volatility
compounds that desorb directly tend to exhibit strongly peaked, Gaussian-like thermograms with
single-mode peaks between around 50 °C to 120 °C; the lower the peak desorption temperature
($T_{peak}$) the higher the volatility of the desorbing compound (Lopez-Hilfiker et al., 2014; 2015). We
therefore refer to thermograms, or portions of thermograms, having this general shape as the
"monomeric" content of the ion hereafter; direct evaporation of thermally stable dimers or other
oligomers is possible, although will typically occur at higher temperatures due to the comparably
lower volatility of these compounds. When multiple monomeric compounds having different
vapor pressures contribute to the same ion, the resulting thermogram exhibits a broader peak
and shallower slopes or, in particular cases, multiple, distinct peaks (Lopez-Hilfiker et al., 2015).
However, very broad thermograms, especially those that peak at higher temperatures (> 120 °C
or so), can also indicate contributions from thermal decomposition of very low-volatility



monomers, dimers, and oligomers (Lopez-Hilfiker et al., 2015; Gaston et al., 2016; Schobesberger
et al., 2018). Dimers and oligomers can evaporate directly, without thermal decomposition, as
observed for isoprene-derived SOA (D'Ambro et al., 2017) and ambient monoterpene oxidation
products (Mohr et al., 2017). However, fragments of dimers or oligomers are generally more
abundant, indicating the importance of thermal decomposition for desorption of these low-
volatility compounds. Both direct evaporation of extremely low-volatility compounds and
decomposition of large molecules or oligomers can lead to high signal levels above ~120 °C. We
refer to both peaks and the slowly varying signal above ~120 °C as the "oligomeric" content of
the ion hereafter. We use the terms monomer and oligomer in a qualitative manner. A more
quantitative analysis of the thermograms can help distinguish between direct evaporation,
thermal decomposition, and the contributions of monomers versus oligomers (Schobesberger et
al., 2018), yet is beyond the scope of the current work.

## 4. Example Applications

To illustrate the broad utility of NSSC for interpretation and analysis of FIGAERO-CIMS data,
we apply NSSC to the laboratory-generated SOA systems described above. The systems include:
SOA formed from a single precursor under $NO_x$-free conditions; SOA formed from a single
precursor as a function of input [NO]; and, SOA formed from a single precursor with thermal
desorption following isothermal evaporation.

### 4.1.    α-pinene + OH SOA

A total of 298 ions were characterized by FIGAERO-CIMS for SOA generated from the
α-pinene + OH reaction (**Table 1**). Four ions were characterized as anomalous and excluded from
further analysis (see Section 2.1.1). The mass concentration of each ion was calculated by
integrating the signal across the entire desorption period and assuming an equal sensitivity of
CIMS for all the compounds. The total mass concentration is the sum of all the non-anomalous
ions. The mass spectrum and bulk thermogram of the remaining 294 ions are shown in **Figure 3**,
with the bulk thermogram shown versus both temperature (**Figure 3**b) and time (**Figure 3**c) to
illustrate the difference between the ramping and soaking periods. The individual thermograms
exhibited a variety of shapes. The noise threshold for this data set was $\xi_{ref}$ = 0.020893. A total of



188 ions were screened out via noise filtering. The remaining 106 ions contribute 92.5% to the
total mass detected by FIGAERO-CIMS. The optimal $\varepsilon$ was established through consideration of
the co-dependencies of $N_c$, $N_{c,total}$, $f_{m,unclustered}$ and $R_{interClst}$ on $\varepsilon$ (**Figure 4**; **Table 3**). For this data
set, we determine the optimal $\varepsilon$ = 2.6. Choice of a much smaller $\varepsilon$, around 1.5, gives a maximum
in $N_c$, but leaves a large fraction of the mass unclustered. Choice of $\varepsilon$ = 2.1 or 2.2 yields larger $N_c$
and $R_{interClst}$ than $\varepsilon$ = 2.6, with a reasonably small $f_{m,unclustered}$. However, there is one type of
thermogram (Clst#11 in **Figure 5**) that is only captured with $\varepsilon \geq 2.6$ and this yields $f_{m,unclustered}$ = 0.
Using $\varepsilon \geq 2.7$ also yields $f_{m,unclustered}$ = 0 and $N_{c,one}$ = 0, but $N_c$ and $R_{interClst}$ decrease from $\varepsilon$ = 2.6,
indicating increasing similarity between clusters with fewer types of shapes captured. The choice
of $\varepsilon$ = 2.6 provides a compromise between maximizing $N_c$, minimizing $f_{m,unclustered}$, and keeping
$R_{interClst}$ above two. The parameters and thresholds used for this data set are summarized in **Table**
**3**.
A total of 11 clusters are identified with no one-member clusters. The unweighted and
mass-weighted average thermograms for each cluster are shown along with the thermograms of
individual members in **Figure 5**a. The differences between weighted and unweighted average
clusters are negligible, in general. Clusters are organized and numbered (as Clst#$N$) from low to
high $T_{m50}$, with deeper to shallower downslope. Clst#1 through Clst#6 all have a clear peak below
120 °C, but with different peak widths and downslopes. Clst#7 and Clst#8 are a bit noisier with
only a few members each, exhibiting a sharp upslope and shallow downslope. Clst#9 has a very
broad peak. Clst#10 peaks at around 150 °C after an initial rise and temporary plateau. Clst#11
exhibits behavior somewhat like Clst#10, but with a peak that occurs just into the soaking period,
evident if viewed in time space, at 200 °C with a rapid drop afterwards.
The total mass concentration of a given cluster ($M_{c,N}$) is the sum across all cluster members,
calculated by integrating the summed mass concentration across the entire desorption period.
The percentage mass contribution of each cluster, and of the unclustered and the noise-filtered
ions, as well as the number of members for each cluster are shown in **Figure 5**b and **Table S1**.
Clst#2 and Clst#3 contain the majority of the mass (20.1% and 44.3%, respectively) and consist
of nearly half of the clustered ions (11 and 42, respectively). Clst#4 and Clst#9 also contain a
notable percentage of the total mass (8.2% and 9.8%, respectively) and include a notable number





of ions (13 and 17, respectively). Other clusters contribute relatively little to the total mass and
contain a small fraction of ions.
The mass-weighted average molecular formulas ($C_xH_yO_zN_m$) differ between clusters, as do
the O:C and H:C atomic ratios (**Table S1**). There is no clear relationship between $T_{m50}$ (or cluster
number) and the number of carbon atoms, MW, or O:C. There is, however, a reasonable, inverse
correlation between $T_{m50}$ and H:C ($r^2 = 0.78$). The number of carbon atoms is notably larger for
Cluster 6 (x = 11.1) and Cluster 7 (x = 15.3); if those two clusters are excluded there is an inverse
relationship between $T_{m50}$ and the number of carbon atoms ($r^2 = 0.79$) and with MW ($r^2 = 0.59$).
While the reason for these two clusters having comparably large numbers of carbon atoms is
unknown, this nonetheless suggests that the contribution of oligomer decomposition might
increase for clusters having higher $T_{m50}$ values.
Interpretation of previous FIGAERO-CIMS studies have largely focused on the behavior of
the bulk thermogram or of several major ions or sums of ions based on common factors such as
the number of carbon atoms (Lopez-Hilfiker et al., 2016; D'Ambro et al., 2017; D'Ambro et al.,
2018; Stolzenburg et al., 2018; Wang and Ruiz, 2018; Joo et al., 2019). The normalized
thermograms of the top five ions contributing most to the total mass for the experiments here
are shown in **Figure 5**c, along with the bulk thermogram. Together these five ions make up nearly
30% of the total mass, and exhibit very similar thermogram shapes to each other and to the bulk
thermogram and belong solely to either Clst#2 or Clst#3. Thus, examining these ions only would
capture only a fraction of the overall diversity in thermal behaviors. The clustering method
developed here provides a means to investigate more comprehensively the variability in volatility
between aerosol components.
**4.2.   Δ-3-carene + OH SOA**
A total of 298 ions were characterized by FIGAERO-CIMS for SOA generated from the
reaction of Δ-3-carene + OH (**Table 1**). Five were identified as having anomalous thermograms
and excluded from further analysis. The mass spectrum and bulk thermograms of Δ-3-carene +
OH SOA are shown in **Figure 6**. Compared to the α-pinene +OH SOA described above, the mass
spectrum of Δ-3-carene SOA is quite different, with one ion ($C_8H_{12}O_5$) dominant. The bulk
thermograms of the two SOA systems both look bell-like, but with the Δ-3-carene SOA





thermogram having a peak temperature ca. 9 °C higher. After noise-filtering, 110 ions remained
for clustering, contributing 90.7% to the total mass. The optimal $\varepsilon$ = 2.1, established again by
considering the system-specific dependence of $N_c$, $N_{c,one}$, $f_{m,unclustered}$ and $R_{interClst}$ on $\varepsilon$ (**Figure S1**),
with the parameters and thresholds summarized in **Table 3**.
Ten clusters are identified, including one one-member cluster, with thermograms shown in
**Figure 7a** and the mass contribution and number of ions in a cluster in **Figure 7b**. Chemical
properties of each cluster are summarized in **Table S2**. The general characteristics of
thermograms identified in the $\Delta$-3-carene + OH SOA are similar to those of low-NO$_x$ $\alpha$-pinene +
OH SOA described above, but with different mass contributions. For example, Clst#4 has nearly
identical shape of the thermogram as Clst#3 in the $\alpha$-pinene SOA but contributes less to the total
mass, 28.0% compared to 44.3%. Clst#6 in the $\Delta$-3-carene SOA contributes 14.8% to the total
mass and resembles Clst#5 in the $\alpha$-pinene SOA, which contributes only 4.0% to the total mass.
In general, Clst#1 – 6 in the $\Delta$-3-carene SOA all exhibit a peak below 120 °C, with clear peaks
of varying width and downslopes of varying steepness, but nominally in order of narrow to wide
and steep to shallow, respectively. These clusters carry the majority of the desorbed mass. Clst#7
and Clst#8 both exhibit relatively flat thermograms in the ramping period after their initial rise,
and contribute 9% to the total mass. Clst#9 has a peak temperature above 150 °C and Clst#10
reaches a maximum during the soaking period. These last two clusters contribute little to the
total mass (0.6% and 0.3%, respectively).
The thermograms of the five largest ions are shown in **Figure 7c**. These five ions together
carry ~35% of the SOA mass. A wider variety of thermogram shapes are captured by the top five
ions compared to the $\alpha$-pinene SOA system. However, thermograms characteristic of Clst#7–10
are not represented by these top five ions; this remains true even if the top 10 ions are
considered (not shown).
There are ultimately three major differences between the two SOA systems. For one, there
is a different relationship between fractional contribution and cluster number (and thus $T_{m,50}$)
between the two. Secondly, the $\alpha$-pinene SOA contains ions with especially narrow peaks at ca.
100 °C (i.e., Clst#7 & 8), that are not observed with $\Delta$-3-carene SOA (compare **Figure 5** with **Figure**
**7**). Lastly, the thermograms of the top five ions for $\Delta$-3-carene SOA differ to a greater extent than



for α-pinene SOA. Although we are unable to determine the reasons for these differences here,
this illustrates the potential for clustering to help identify and understand differences between
different SOA systems.

### 4.3.    α-pinene + OH + NO SOA

Thermograms from SOA generated from the reaction of α-pinene + OH at varying NO
concentrations (5 ppb, 10 ppb and 25 ppb; **Table 1**) are considered as a set of experiments.
Together, differences between them illustrate the impact of changes to the fate of $RO_2$ peroxy
radical intermediates on the SOA composition and thermal properties (Praske et al., 2018; Zhao
et al., 2018). Clustering proceeds here using two complementary approaches. In the single
clustering method, clustering is performed for one reference experiment (i.e., at one NO
concentration, 5 ppb, Expt#3a). Then, average thermograms are calculated for the other
experiments in the set using the same cluster members as identified in the reference experiment.
In the multiple clustering method, clusters are independently determined for each experiment in
the set, and the shapes, relative abundances, and contributing ions are compared between
experiments. For all three experiments, the same initial set of 298 ions were characterized by
FIGAERO-CIMS.

### 4.3.1.  Single Clustering

The ions identified as anomalous in each experiment differed. This most likely results from
shifts in the background signal levels between experiments. To maintain consistency between
the three experiments, ions identified as anomalous in any of the experiments were excluded
from all the experiments, with four ions excluded in total. A total of 88 ions were kept for
clustering after noise-filtering using the 5 ppb NO reference experiment, contributing 84.5% to
the total mass. The optimal ε = 2.2 (**Figure S2** and **Table 3**), resulting in ten clusters with one
one-member cluster. The same sets of ions were then used to calculate the cluster-average
thermograms for the 10 ppb and 25 ppb NO experiments. Chemical characteristics of the clusters
are summarized in **Table S3**.
Mass spectra for the three experiments are compared in **Figure 8**a and the bulk
thermograms shown in **Figure 8**b and c. The 5 ppb NO and 10 ppb NO SOA mass spectra are





nearly identical. The mass spectrum for the 25 ppb NO experiment, however, exhibits a notable
shift of the most abundant ions towards lower m/z. The bulk thermograms for the 5 ppb and 10
ppb NO experiments are nearly identical, peaking near 80 °C. The 25 ppb NO bulk thermogram
similarly peaks near 80 °C, but exhibits a much slower decay as temperature increases further.
Additionally, the change in slope at the transition from the ramping to soaking period is more
pronounced in the 25 ppb NO experiment. Overall, a greater fraction of the mass desorbs above
100 °C and during the soaking period for the 25 ppb NO experiment compared to lower-NO
experiments.

Despite the differences in the bulk thermograms, the shapes of the weighted-average

thermograms of clusters for all the NO experiments are generally similar, with the exception of
Clst#6 (**Figure 9**a). In particular, the 25 ppb thermogram shape of Clst#6 differs substantially from
those of low-NO conditions, with a much reduced initial peak (around 80 °C) and an more
pronounced second peak at high temperature (around 200 °C). However, this cluster contributes
negligibly to the overall mass. There is some suggestion of similar behavior for Clst#10, although
to a lesser extent. For the three most abundant clusters, Clst#1, 2 and 4, there is a slightly
increased relative contribution of the 100-200 °C tail for 25 ppb NO, consistent with differences
in the bulk thermograms.

The most notable NO-dependent change is in the relative abundances of the clusters

between the 5 and 10 ppb NO experiments and the 25 ppb NO experiment (**Figure 9**b). The
cluster mass fractions are nearly identical between the 5 and 10 ppb NO experiments. The
relative contributions of higher-number clusters (which have been ordered according to
increasing $T_{m,50}$) increase for the 25 ppb NO experiment. This is consistent with the increased
persistence of the 25 ppb NO bulk thermogram to higher temperatures and the nearly identical
nature of the 5 ppb and 10 ppb NO bulk thermograms (**Figure 8**b). The clustering analysis suggests
that differences in the bulk thermogram arise from shifts in the relative contributions of the
various SOA components that result from the altered photochemical environment.   These
observations generally suggest an increasing fraction of oligomeric content, or less-volatile
compounds, formed in the particle phase—or potentially the gas phase—when the SOA was
generated under higher chamber NO conditions (Schobesberger et al., 2018).



### 4.3.2. Multiple Clustering


With multiple clustering, each experiment was processed and clustered independently,
with experiment-specific $\xi_{ref}$, $N_c$, and $\varepsilon$, among other parameters (**Figure S4** and **Table 3**). The
clustered thermograms from the three experiments are compared in **Figure 10**a-c. The number
of clusters identified increases with NO concentration. Comparison between the shapes of the
clusters from the 5 ppb NO (**Figure 10**a) and 10 ppb NO (**Figure 10**b) experiments indicates
generally similar types of thermograms, consistent with the single clustering method. Ten of the
11 total 10 ppb clusters match with a 5 ppb cluster. The one additional, unique cluster at 10 ppb
NO (Clst#9), is a one-member cluster with a sharp, narrow peak at low temperatures and a
broader, shallow second peak at high temperatures. This ion was filtered out due to high noise
level in the 5 ppb NO experiment.
The 25 ppb NO experiment (**Figure 10**c) results in more clusters compared to the lower NO
experiments; 13 for the 25 ppb NO experiment versus 10 and 11 for the 5 and 10 ppb experiments,
respectively. Some of the 25 ppb NO clusters have shapes similar to the lower NO experiments,
but many differ substantially. For example, two of the unique 25 ppb NO clusters (Clst#12 and
#13) have thermograms for which the signal increases continuously through the ramping period
and even into the soaking period. These clusters were not found in the single clustering analysis
because the 5 ppb NO experiment was used as the reference.
The new types of thermograms observed in the 25 ppb NO experiment indicates either
formation of new compounds or a change in the relative contributions of different components
to the same ions. Either could result from a change in the fate of the peroxy radical intermediates
as the NO concentration increases, leading to notably different products. There were numerous
nitrogen-containing ions observed for the three experiments. These N-containing ions belong to
Clst#1 – 7 for all the three [NO] conditions (**Table S4**). The higher-number clusters did not include
N-containing ions, also indicating a limited influence of the N-containing products on these lower-
volatility thermograms, although fragmentation complicates the interpretation. Overall, the
formation of new N-containing compounds at the high NO condition does not seem to explain
the unique thermograms in the 25 ppb NO experiments.



The percent contribution of different clusters to total mass, along with the noise-filtered
and unclustered ions, differ between experiments (**Figure 10**d). Note that for the multiple
clustering method, clusters having the same index number are not necessarily directly
comparable between experiments because different sets of ions are included. For example, while
Clst#1 in the 5 ppb and 10 ppb NO experiments are comparable, the most similar cluster in the
25 ppb experiment is Clst#2. Nonetheless, there are some common features shared by the same,
or closely indexed, clusters. For example, Clst#1 – 4 in all three experiments exhibit a narrow,
single peak with the peak temperature below 120 °C. The mass contribution of Clst#1 – 4 is similar
between the 5 and 10 ppb NO experiment, but ~15% lower in the 25 ppb NO experiment. Clusters
that reach their maximum signal at or above 150 °C (Clst#9, 10 for 5 ppb, Clst#10, 11 for 10 ppb
and Clst#10 – 13 for 25 ppb) together contribute ~6% in the low NO experiments and ~13% in
the high NO experiments. Thus, there is some evidence that at higher NO there is an increased
contribution of oligomeric compounds, indicated by the increased contribution of clusters that
peak at higher temperatures and exhibit broader overall thermograms. However, overall these
observations suggest complex shifts in the distribution of products, both monomeric and
oligomeric, with sufficient increases in NO to change the fate of the peroxy radical intermediates.
### 4.4.  α-pinene + O$_3$ SOA
SOA formed from dark ozonolysis of α-pinene was collected and then allowed to
isothermally evaporate for varying amounts of time (0 h, 1 h, 3 h, 6 h and 24 h) before thermal
desorption (**Table 1**, Expt#4). As above for the SOA formed at varying NO concentrations, these
experiments are considered as a set and interpreted using both the single-clustering and
multiple-clustering approaches. The single-clustering approach uses the 0 h (no-wait) experiment
as the reference for initial clustering. In this set of experiments, 312 ions were characterized by
FIGAERO-CIMS for each experiment.
### 4.4.1.  Single Clustering
Only a few ions, if any, were identified as anomalous in each experiment; a total of ten ions
were removed from all the experiments to maintain consistency between experiments. The mass
spectra and bulk thermograms of the remaining 302 ions for the five experiments are shown in



**Figure 11**. As the isothermal evaporation time increases, the mass spectrum changes significantly,
as previously reported by D'Ambro et al. (2018). In the no-wait experiment, the mass spectrum
is dominated by one ion, $C_{10}H_{14}O_6$. Upon isothermal evaporation, the relative abundance of this
ion notably decreases, with the extent of decrease increasing with wait time; over time, a greater
number of ions contribute to the total mass, both at lower and higher $m/z$. With isothermal
evaporation, the bulk thermograms also exhibit a shift from a more peaked shape, reminiscent
of that from a single compound (Lopez-Hilfiker et al., 2014), to a more flattened peak with a
shallower rise (**Figure 11**). In other words, with increasing isothermal evaporation the majority
of the mass desorbed during thermal desorption shifts from a lower to higher temperature region.
This behavior largely reflects the loss of comparably more volatile compounds during isothermal
evaporation, leaving behind SOA that is overall less volatile (**Figure S6**a). It can also in part be due
to higher molecular weight, lower volatility compounds being produced with time via accretion
reactions in the condensed phase.

There are 12 clusters determined from the no-wait experiment, exhibiting a wide variety of

the shapes (**Figure 12**a), with the parameters used for data pre-processing and clustering
reported in **Table 3** and shown in **Figure S5**. Focusing first on the no-wait experiment, the cluster
thermogram shapes include those having clear peaks at relatively low temperatures (~60 °C) and
with a sharp rise and fall (e.g., Clst#1-3), those having sharp peaks at relatively low temperatures
but with a shallow downward slope (e.g., Clst#6), those with a broad peak at somewhat higher
temperatures (~100 °C) and long tails (e.g., Clst#7), and those having a wide peak at even higher
temperatures ~120 °C with a very broad rise and fall (e.g., Clst#10).

Changes to the shapes of the thermograms that occur upon isothermal evaporation differ

between the clusters. Some of the clusters exhibit almost step changes from the no-wait to the
longer time experiments (e.g., Clst#2 and 6), while others exhibit more continuous changes (e.g.,
Clst#3 and 5). However, in all cases the clusters shift to have peaks that occur at higher
temperatures with generally broader thermograms. In other words, the $T_{m50}$ of all the clusters
increase as a function of evaporation time, but with larger increases observed for the clusters
having initially lower $T_{m,50}$ (**Figure 12**b). For some of the clusters with a clear peak below 100 °C,
such as Clst#1–6, the peaks broaden to become less obvious and shift to higher temperatures



with longer isothermal evaporation. For clusters that originally have very wide peaks, such as
Clst#8–10 and 12, isothermal evaporation engenders a general shift in the thermograms towards
higher temperatures. Different from the clusters described above, thermograms for two clusters,
Clst#7 and Clst#11, exhibit only minor shift of peak temperature and shapes. Thermograms of
these two clusters share the common features of a moderate-width peak that reaches a
maximum between 100 – 120 °C. The $T_{m50}$ of these two clusters correspondingly exhibit small
changes compared to other clusters.

Isothermal evaporation generally leads to a reduction of the monomeric character of

clusters, leaving behind components that exhibit increased oligomeric content. Differences in
how the individual cluster thermograms evolve with isothermal evaporation are therefore likely
indicative of differing relative contributions of monomeric versus oligomeric components. For
example, Clst#1 and Clst#10 have distinctly different shapes in the 0-h wait experiment, but very
similar shapes in the 24-h wait experiment. This indicates that ions in Clst#1 are not contributed
from a single component, as might be inferred from the single-mode peak in the 0-h wait
experiment. Instead, they are contributed by multiple components, though initially dominated
by monomeric compounds, so the shift in peak temperature and broadness is substantial. On the
other hand, ions in Clst#10 must also derive from multiple components, but with only a small
fraction of monomeric compounds that evaporate in the 24 hours. Consequently, the loss of
low-temperature mass is apparent yet small. In contrast, ions in clusters such as Clst#7 and 11
must be composed of only low-volatility components because they exhibit minimal changes in
the thermograms shapes.

The extent of mass loss with isothermal evaporation differs between clusters. In general,

clusters that exhibit larger changes in shape have greater total mass loss, although with variability
(**Figure S6**c). Consequently, the mass contributions of the clusters evolve with isothermal
evaporation (**Figure 12**b). The contribution of Clst#1 decreases significantly and most notably as
wait time increases. The most prominent ion in the no-wait experiment, $C_{10}H_{14}O_6$, is grouped in
Clst#1. The continuous mass loss of Clst#1 indicates the rapid evaporation of its members. The
mass contributions of the other clusters that exhibited similar changes in shape as Clst#1 (Clst#3,
5, and 6) remain comparably constant, although with Clst#3 decreasing slightly. The relative





abundances of the clusters for which the thermograms shapes changed negligibly (Clst#7 and 11)
increase continually, implying of the slowest evaporation of the ions in these two clusters in the
24-hr evaporation period.

For comparison, D'Ambro et al. (2018) reported changes in the shapes of the thermograms

for the five most abundant individual ions from the no-wait to 24-hr experiment, together
carrying ~15% of the particle mass. They observed the individual ion thermograms generally all
evolved in a manner similar to our Clst#1, 3 and 5, shifting from narrower, more peaked profiles
towards broader profiles with a shallower rise, less evident peak, and increased evaporation at
higher temperatures. Here, with the clustering of data, we are able to track the change of thermal
behaviors of ions carrying ~87% of the initial mass. We are able to confirm that ~70 % of the mass
exhibit similar thermal behaviors and responses to isothermal evaporation as the top five ions.
However, we are also able to identify another ~17% of the mass having initial thermograms not
characterized by the top five ions, including 12% of the mass (Clst#7 and 11) that behaves
distinctly different upon evaporation at room temperature.
**4.4.2.  Multiple Clustering**

The number of clusters identified with the multiple-clustering method, using experiment-

specific optimal $\varepsilon$ values (**Table 3** and **Figure S7**), decreases with isothermal evaporation time,
from 13 (no-wait) to 12 (1 h) to 11 (3 h) and then to 9 (6 h and 24 h) (**Figure 13**b-f). The noise
levels of the thermograms increase with evaporation time due to decreasing absolute particle
mass. Nonetheless, the typical shapes of the cluster-specific thermograms clearly evolve with
increasing isothermal evaporation. For short isothermal evaporation times, many cluster-specific
thermogram profiles are relatively narrow, peaking at lower temperatures (70-120 °C) and with
rapid rises and evident downslopes. For longer isothermal evaporation times, the cluster-specific
profiles instead have broad peaks with slow rises and most of the mass desorbing at higher
temperatures.

To aid further general interpretation, the cluster-specific thermograms with $T_{m50} < 120$ °C

are grouped together as higher-volatility clusters. The number of higher-volatility clusters
decreases with isothermal evaporation, from ten for the no-wait experiment, to five in the 1-h





experiment, two in the 3-h and 6-h experiment, to none in the 24-h experiment (**Figure 14**). The
mass contributions of the higher-volatility clusters decrease from 81.9% to 60.4%, 17.2%, 9.4%
and to 0.0%, with increasing isothermal evaporation time. This overall behavior is consistent with
results from the single-clustering method and indicates the compounds with a wide range of
volatilities make up much of the mass in the initial particles, while the SOA after isothermal
evaporation is composed of compounds having lower volatilities.

After isothermal evaporation, some cluster-specific thermograms have signals that increase

continuously during the ramping period, for example Clst#11 and 12 in the 1-h experiment; such
clusters were not observed in the no-wait experiment. The relative abundance of these very low-
volatility clusters increases with isothermal evaporation, from 1.7% in the 1-h experiment
(Clst#11 and 12) to 13.4% in the 24-hr experiment (Clst#7 and 9). The absence of these clusters
for the no-wait experiment suggests that they are formed over time through condensed-phase
reactions. Their increasing contribution over time may reflect both evaporation of higher
volatility components and continued formation. Clusters having thermograms with very broad
peaks, such as Clst#11 and 13 in the 0-h experiment are also observed in all the other experiments,
with increasing contribution to the total mass.

The multiple-clustering method reveals the disappearance of certain types of thermograms,

(e.g., the no-wait Clst#3) and the emergence of other types of thermograms (e.g., the 1-h Clst#11)
as evaporation time increases. This complements the single-clustering method, which illustrates
gradual changes in the shapes of cluster-specific thermograms, by allowing for identification of
completely new thermogram shapes and divergent behavior between ions within initial clusters.
The multiple-clustering method also confirms the decrease of the diversity of the desorption
profiles, as suggested by the single-clustering method. The two methods complement each other
and together provide a detailed look into (i) how the desorption profiles of sets of ions evolve
with isothermal evaporation and (ii) how the fraction of different types of thermograms change
with evaporation time.



## 5. Conclusions

We developed a new clustering algorithm, the noise-sorted scanning clustering (NSSC) algorithm, for application to FIGAERO-CIMS data sets. The NSSC algorithm provides a robust method for clustering of FIGAERO-CIMS thermograms having distinct thermal desorption profiles and of determining the mass contribution of each cluster. Each of the ions contributing to a cluster results from one or more molecules sharing similar thermochemical properties. These molecules either evaporate directly or decompose and then evaporate. Compared to other existing clustering algorithms, NSSC is strictly similarity-based, reproducible, and takes into consideration differences in noise levels between individual ions. The application of NSSC has the potential to make FIGAERO data more accessible to the atmospheric chemistry community.

For the four different SOA systems we examined, more than 80% of the total mass is clustered, with the number of clusters ranging from 9 to 13. The shapes of the cluster-specific average thermograms exhibit substantial variation for a given system. Some have relatively sharp peaks, others broad peaks with slowly decreasing signal as heating continues, and others still having signals that continually increase up to very high temperatures or long desorption times. The mass contribution of a cluster varies from 0.2% to 44.3%. A few (2-3) clusters usually contain more than 50% of the total mass in all the chemical systems examined. Comparison of the cluster-specific thermogram shapes between different SOA systems allows for qualitative assessment of the similarity or uniqueness.

We also demonstrated the potential of the NSSC for guiding interpretation of sets of experiments where one experimental condition varies (e.g., NO concentration and evaporation time). For such experiments, two complementary methods are suggested: (i) the single clustering method, where one experiment is used to determine the ions belonging to individual clusters and then clusters comprising the same ions are calculated for the other experiments, and (ii) the multiple clustering method, where each experiment is clustered independently and then compared. The first approach helps establish how the properties of individual clusters evolve as a set, while the second approach helps identify changes in the diversity of cluster-specific thermogram shapes, properties, and mass contributions. The two approaches complement each





other and provide guidance for future efforts to cluster ambient observations having long time-
series.
This paper focuses only on the description of the clustering algorithm and its potential as a
tool to characterize the properties of organic aerosol in further detail. Interpretation of the
cluster-specific thermograms using frameworks such as that of Schobesberger et al. (2018) will
allow for more comprehensive understanding of the thermochemical properties of the organic
aerosol, the subject of future work. This will provide insights into the thermal behavior of organic
aerosol and the relative contributions of thermally stable (e.g., monomer) versus thermally
unstable (e.g., dimers or oligomers) compounds, the volatility distribution of the thermally stable
compounds, and the T-dependent rate coefficients for oligomer dissociation and formation.

## 6. Data Availability


All data and the NSSC algorithm used in this publication are archived in the UC DASH data
repository (Cappa et al., 2019). The NSSC algorithm is also available at GitHub
([https://github.com/chriscappa/NSSC](https://github.com/chriscappa/NSSC)), with the version used for this publication available as Li
and Cappa (2019).

## 7. Author Contributions


ZL developed the NSSC algorithm. ELD, SS, CJG, FDL-H, JL, JES, and ZL performed
measurements. ELD and SS performed detailed data processing. ZL and CDC analyzed data and
wrote the manuscript, with contributions from all co-authors.

## 8. Acknowledgements


This work was supported by the National Science Foundation under Grant No. ATM-
1151062. The experimental work described here was supported by the U.S. Department of
Energy ASR grants DE-SC0011791 and DE-SC0018221. E.L.D. was supported by the National
Science Foundation Graduate Research Fellowship (grant no. DGE-1256082) and S.S. was
supported by the Academy of Finland (grant nos. 272041 and 310682). The SOAFFEE campaign
was done at Pacific Northwest National Laboratory, supported by the U.S. Department of Energy
(DOE) Office of Science, Office of Biological and Environmental Research, as part of the



Atmospheric Systems Research (ASR) program. PNNL is operated for DOE by Battelle Memorial
Institute under contract DE-AC05-76RL01830.

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




## 10.    Tables

**Table 1.** Details of SOA formation and chamber conditions for all the example SOA systems.

| Exp # | Precursor | | Oxidant | | Seeds | | UV | T (°C) | RH ( %) | NO[#$] (ppb) | $M_p$[#&] (μg/m³) | FIGAERO Operation [$] |
|---|---|---|---|---|---|---|---|---|---|---|---|---|
| | Type | Conc.[#] (ppb) | Type | Conc.[##] (ppm) | Type | $D_p$[#*] (nm) | | | | | | |
| 1* | α-pinene | 10 | OH (H$_2$O$_2$) | 1.0 | AS[&] | 50 | On | 25 | 50 | - | 5.1 | Normal |
| 2 | Δ-3-carene | 10 | OH (H$_2$O$_2$) | 0.25 | AS | 50 | On | 25 | 50 | - | 5.2 | Normal |
| 3a | α-pinene | 10 | OH (H$_2$O$_2$) | 1.0 | AS | 50 | On | 25 | 50 | 5 | 8.3 | Normal |
| 3b | | | | | | | | | | 10 | 9.2 | |
| 3c | | | | | | | | | | 25 | 9.1 | |
| 4a | α-pinene | 10 | O$_3$ | 0.1 | PS[&&] | 50 | Off | 25 | 80 | - | 4.0 | Normal |
| 4b | | | | | | | | | | | | 1 h wait |
| 4c | | | | | | | | | | | | 3 h wait |
| 4d | | | | | | | | | | | | 6 h wait |
| 4e | | | | | | | | | | | | 24 h wait |

\* Experiment #1 is a case study used to test the performances of different clustering algorithms

[#] Conc. of precursors are the concentrations expected in the chamber with the absence of any chemistry

[##] For OH, conc. refers to concentration of H$_2$O$_2$ injected into the chamber; for O$_3$, conc. refers to steady-state concentration of O$_3$ in the chamber during SOA formation

[#*] Seed particles are size-selected in all the experiments

[#$] NO concentration refers to the targeted NO concentration when NO is injected into the chamber. The actual steady-state concentration of NO is lower than targeted. "-" indicates that no external NO is added to the chamber

[#&] $M_p$ is the estimated mass concentration of particles including SOA and seeds measured by SMPS when the chamber is at steady-state, except for experiment 4 where $M_p$ is the mass concentration of SOA only

[$] Normal operation mode means the desorption process starts immediately after collection period. X h wait means that particles are isothermally diluted for X hours before the desorption process is initiated

[&] AS = ammonium sulfate

[&&] PS = potassium sulfate







**Table 2.** Comparison of different clustering algorithms

| Clustering Algorithms | k-means | k-medoids | Mean-shift | DBSCAN | FPClustering | NSSC |
|---|---|---|---|---|---|---|
| Assign all the members? | Yes | Yes | No | No | Yes | No |
| Identify single-member clusters? | No | No | Yes | No | No | Yes |
| Robust solution? | No | No | No | Yes | No | Yes |
| Controlled distance from the center of clusters? | No | No | Yes | No | No | Yes |
| Influence of noise? | large | large | small | small | large | Small |
| Key preset parameters | $N_c$ | $N_c$ | $\varepsilon, N_{min}$ | $\varepsilon$ | Initial seed | $\varepsilon, N_{min}$ |
| Software used in this study | Igor | R | Python | Igor | Igor | Igor |






**Table 3**. Parameters and thresholds used for the data processing and noise-sorted scanning clustering for
all the example experiments.

| Expt # | SOA type | | Pre-processing | | | | | | Clustering | | | | |
|---|---|---|---|---|---|---|---|---|---|---|---|---|---|
| | | | $N_{total}$ | $N_{anomalous}$ | $N_{filtered}$ | $f_{m,filtered}$ | $\xi_{ref}$ | $f_{m,ref}$ | $\varepsilon$ | $N_c$ | $N_{c,one}$ | $f_{m,unclustered}$ | $R_{interClst}$ |
| 1 | α-pinene + OH | | 298 | 4 | 188 | 7.5 | 0.021 | 0.67 | 2.6 | 11 | 0 | 0.00 | 2.01 |
| 2 | Δ-3-carene + OH | | 298 | 5 | 183 | 9.3 | 0.019 | 0.57 | 2.1 | 9 | 1 | 0.27 | 2.36 |
| 3a | | Single | 298 | 6 | 204 | 15.3 | 0.025 | 0.55 | 2.2 | 9 | 1 | 1.52 | 2.06 |
| 3b | | | | 6 | 204 | 17.5 | - | - | - | 9 | 1 | 1.72 | - |
| 3c | α-pinene + OH + NO | | | 6 | 204 | 21.0 | - | - | - | 9 | 1 | 2.27 | - |
| 3a | | Multi | 298 | 2 | 208 | 15.5 | 0.025 | 0.55 | 2.2 | 9 | 1 | 1.52 | 2.06 |
| 3b | | | | 3 | 195 | 12.6 | 0.027 | 0.54 | 2.3 | 10 | 1 | 1.29 | 2.10 |
| 3c | | | | 6 | 200 | 12.8 | 0.028 | 0.43 | 2.5 | 12 | 1 | 1.21 | 1.96 |
| 4a | | Single | 312 | 10 | 185 | 11.5 | 0.025 | 0.42 | 2.2 | 10 | 2 | 0.67 | 2.28 |
| 4b | | | | 10 | 185 | 14.0 | - | - | - | 10 | 2 | 0.79 | - |
| 4c | | | | 10 | 185 | 14.0 | - | - | - | 10 | 2 | 0.84 | - |
| 4d | | | | 10 | 185 | 13.8 | - | - | - | 10 | 2 | 0.83 | - |
| 4e | α-pinene + O₃ | | | 10 | 185 | 17.6 | - | - | - | 10 | 2 | 0.82 | - |
| 4a | | Multi | 312 | 1 | 191 | 11.4 | 0.025 | 0.41 | 2.2 | 11 | 2 | 1.04 | 2.22 |
| 4b | | | | 0 | 210 | 16.5 | 0.044 | 0.41 | 3.3 | 8 | 4 | 0.00 | 2.02 |
| 4c | | | | 5 | 205 | 14.3 | 0.048 | 0.42 | 3.1 | 9 | 2 | 1.06 | 1.66 |
| 4d | | | | 3 | 203 | 12.8 | 0.055 | 0.39 | 3.3 | 8 | 1 | 2.50 | 1.80 |
| 4e | | | | 3 | 213 | 16.1 | 0.053 | 0.41 | 3.4 | 7 | 2 | 0.98 | 1.97 |

$N_{total}$ – Total number of ions characterized by CIMS

$N_{anomalous}$ – Number of anomalous ions

$N_{filtered}$ – Number of ions filtered out from the following clustering due to high levels of noises

$f_{m,filtered}$ – Mass fraction of the ions filtered out due to high levels of noises, expressed in %

$\xi_{ref}$ – Noise threshold. Ions with noise levels above this threshold are excluded from clustering

$f_{m,ref}$ – The threshold of mass contribution (%) to identify an ion as significant

$\varepsilon$ – distance criterion

$N_c$ – Number of clusters determined with two or more members

$N_{c,one}$ – Number of clusters determined with only one member

$f_{m,unclustered}$ – Mass fraction of unclustered ions, expressed in %

$R_{interClst}$ – The ratio of the average inter-cluster distance over the distance criterion $\varepsilon$






**11.        Figures**

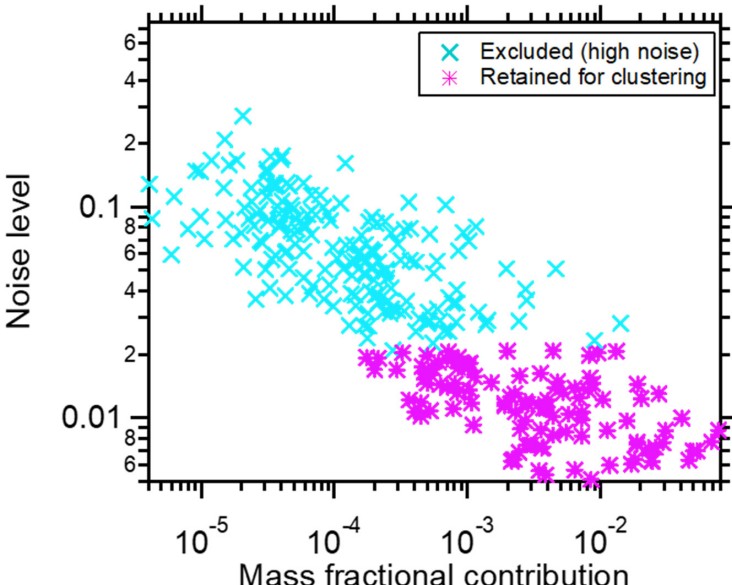


**Figure 1**: The relationship between thermogram noise levels and the fractional contributions of the
corresponding ions to total mass, for α-pinene + OH SOA. The noise threshold, $\xi_{ref}$ = 0.021 and is used to
distinguish high-noise thermograms (cyan markers) from thermograms having acceptable noise levels
(pink markers).

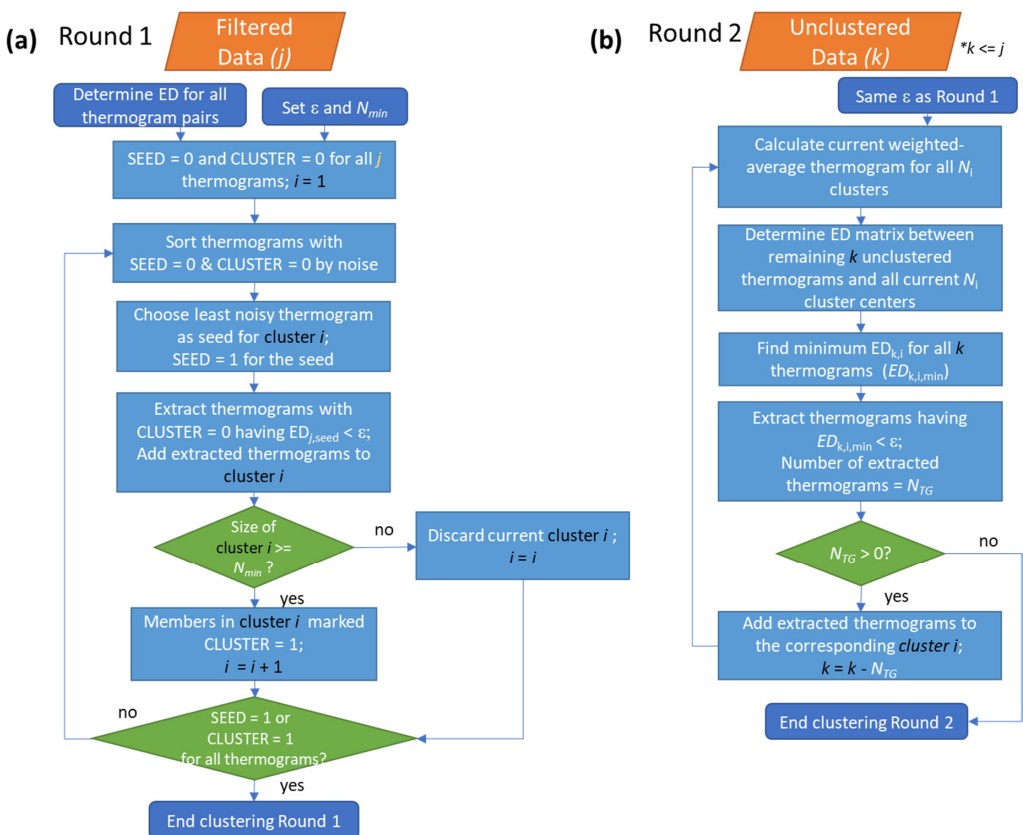


**Figure 2:** Flow of the noise-sorted scanning clustering. There are two rounds of clustering. (a) Round 1: The ED between all thermogram pairs are calculated and two parameters, $\varepsilon$ and $N_{min}$, are set. Each thermogram is initialized with state SEED = 0 and CLUSTER = 0. Only thermograms with SEED = 0 and CLUSTER = 0 can serve as seeds, while thermograms with CLUSTER = 0 can be added to new clusters. The procedure terminates when all the thermograms are marked either SEED = 1 or CLUSTER = 1. (b) Round 2: Seeds are specified as the weighted-average thermogram for each cluster, and any remaining unclustered thermograms from Round 1 are potentially added to these clusters. With the indexing, $j$ refers to the total number of thermograms, $i$ to the number of clusters, and $k$ to the number of unclustered thermograms after Round 1.




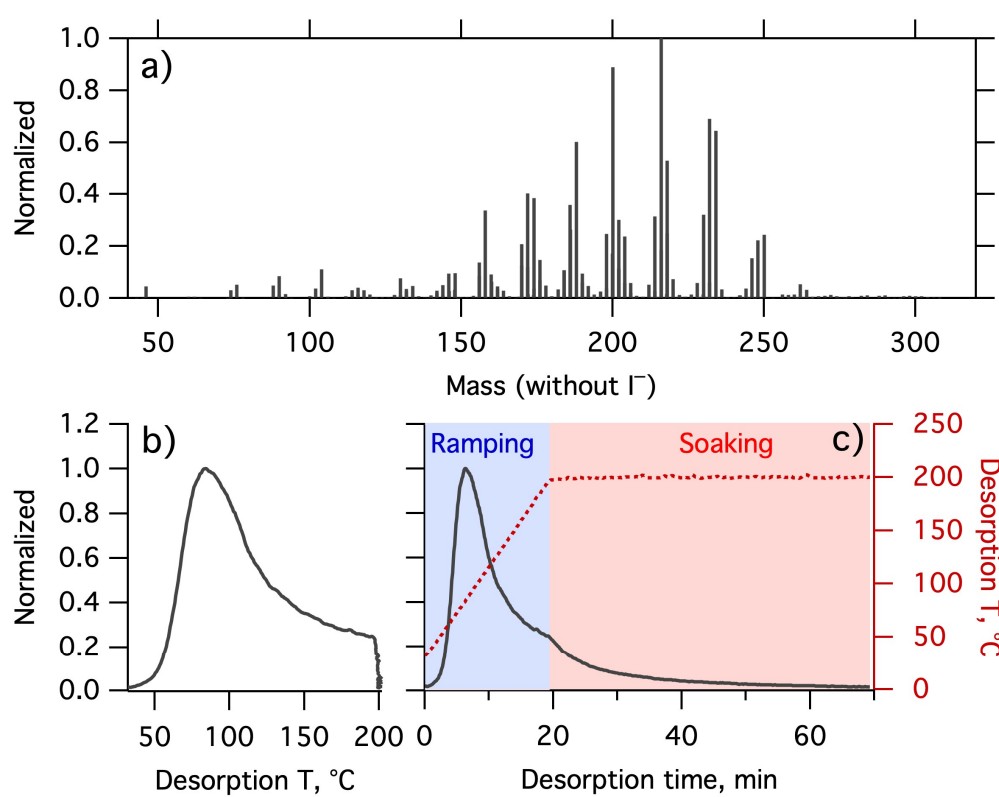

**Figure 3.** (a) Mass spectrum of α-pinene + OH SOA measured by FIGAERO-CIMS. The mass excludes iodine.
(b) Normalized thermogram of the bulk SOA versus temperature. (c) Normalized thermogram of the bulk
SOA versus time (black line) and the variation in desorption temperature with time (dark red dashed line).
The long tail during the soaking period is evident when the thermogram is considered in time space. The
light blue shaded area denotes the ramping period and the pink shaded area the soaking period.






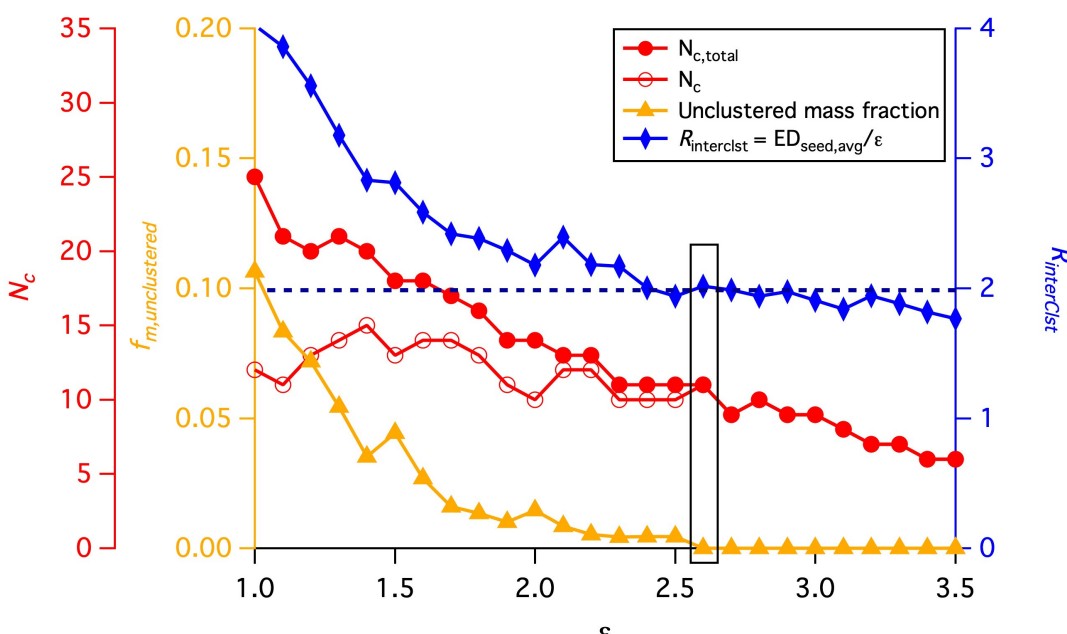

**Figure 4.** The variation of four parameters, $N_c$, $N_{c,total}$, $f_{m,unclustered}$ and $R_{interClst}$ as a function of the distance
criterion $\varepsilon$. The black horizontal dashed line guides the judgement for $R_{interClst} \geq 2$. The values highlighted
by a rectangle are the values corresponding to the optimal $\varepsilon$ used for the clustering analysis.





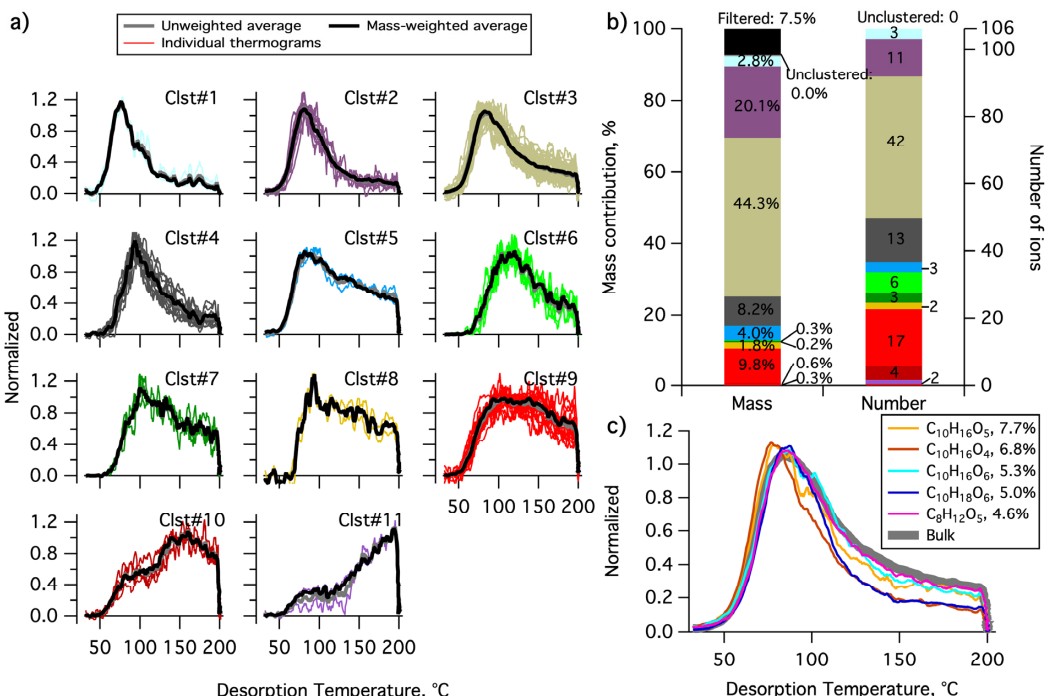


**Figure 5.** Clustering results for α-pinene + OH SOA. (a) Unweighted average thermograms (bold grey lines),
mass-weighted average thermograms (bold black lines) and individual members (colored lines) of the 11
clusters identified. (b) Percentage contribution of each cluster to the total mass, as well as the filtered out
and unclustered mass percentage (left bar), and the number of ions in each cluster and the unclustered
number of ions (right bar). (c) Thermograms of the top 5 ions in terms of mass contribution. The cluster
colors are consistent between (a) and (b).







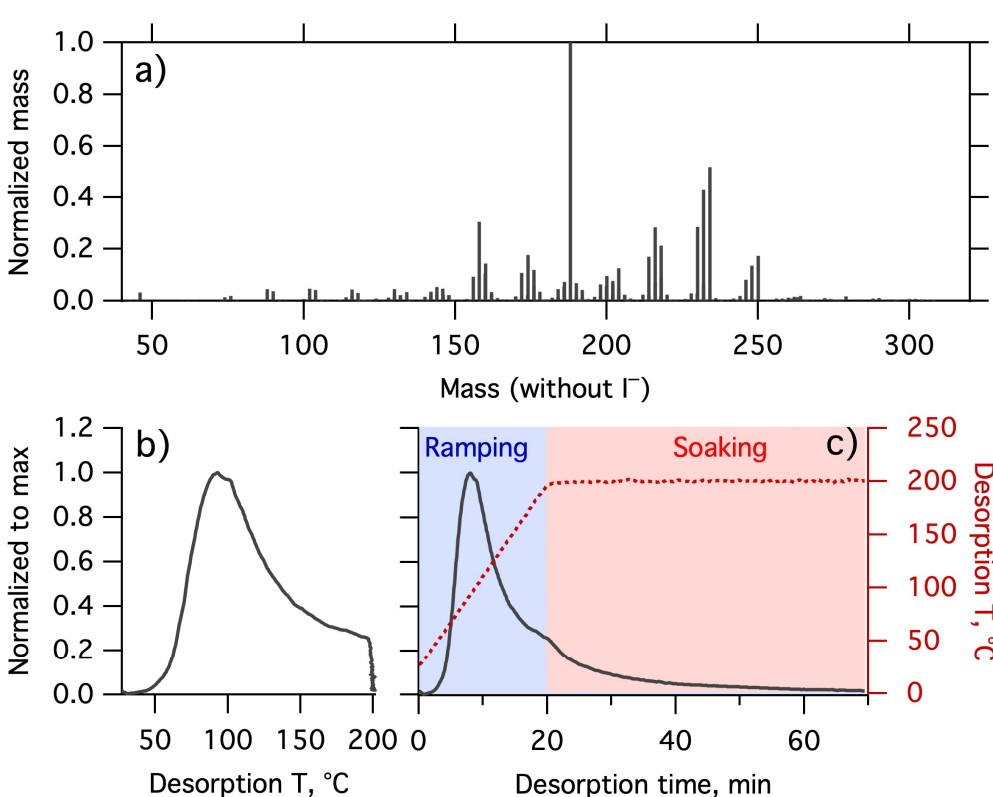

**Figure 6.** Same as Figure 3, but for Δ-3-carene + OH SOA. (a) SOA mass spectrum measured by
FIGAERO-CIMS. The mass excludes iodine. The normalized thermogram of the bulk SOA versus (b)
temperature and (c) time. In (c) the light blue shaded area denotes the ramping period and the pink
shaded area the soaking period.

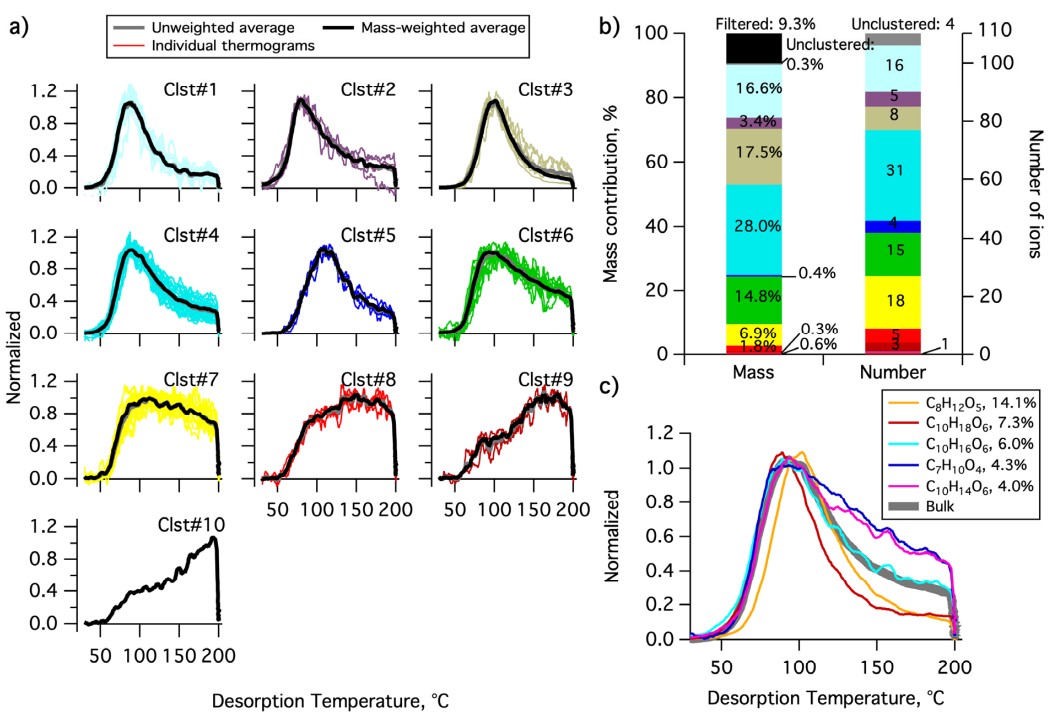

**Figure 7.** Same as Figure 5, but for Δ-3-carene + OH SOA. (a) Unweighted average thermograms (bold grey
lines), mass-weighted average thermograms (bold black lines) and individual members (colored lines) of
the ten clusters identified. (b) Percentage contribution of each cluster to the total mass, as well as the
filtered out and unclustered mass percentage (left bar) and number of ions in each cluster and the
unclustered number of ions (right bar). (c) Thermograms of the top 5 ions in terms of mass contribution.
The cluster colors are consistent between (a) and (b).

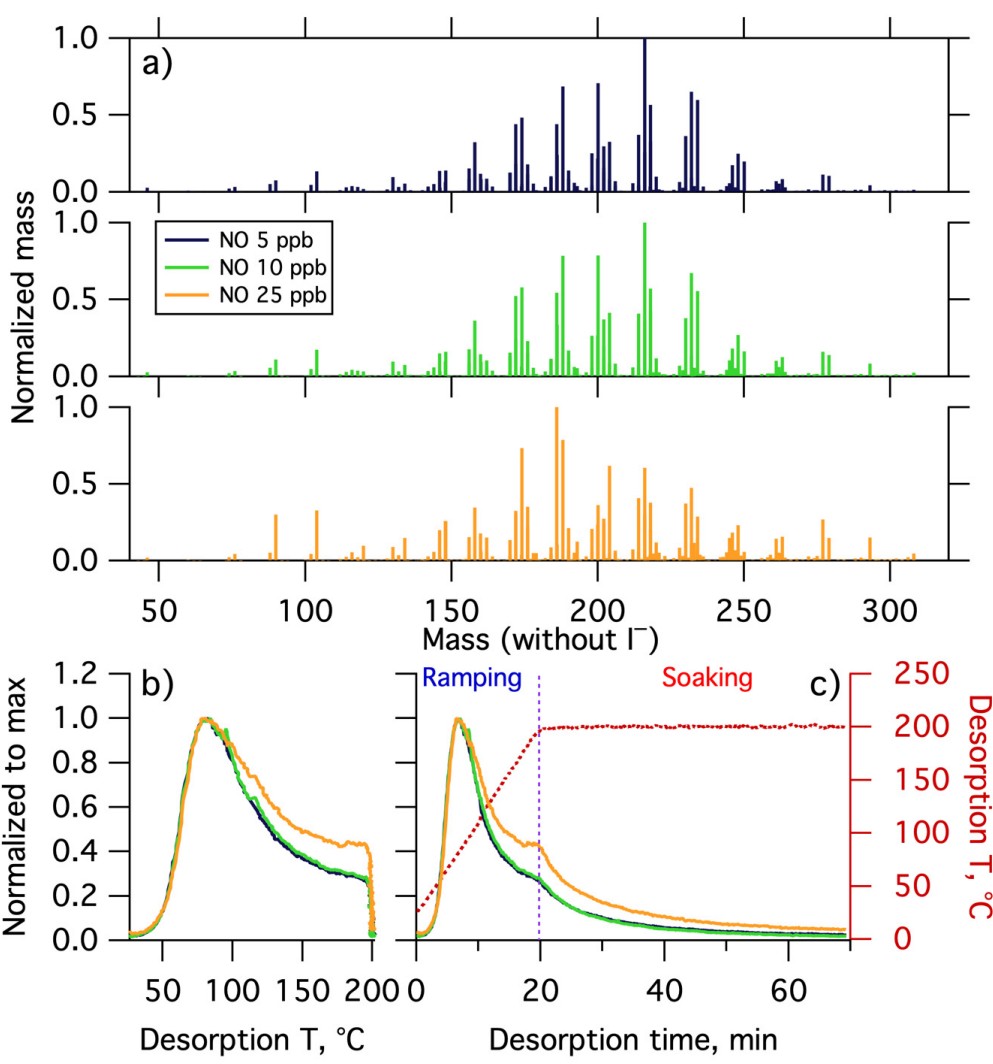

**Figure 8.** (a) Mass spectra of α-pinene + OH SOA formed with different NO concentrations, normalized to
the most abundant ions mass concentration. The mass excludes iodine. Normalized thermograms of the
bulk SOA versus (b) temperature and (c) desorption time, with the desorption temperature shown in dark
red dashed line. The vertical purple dashed line delineates between ramping and soaking. In all the panels,
colors correspond to the NO concentration (see legend).



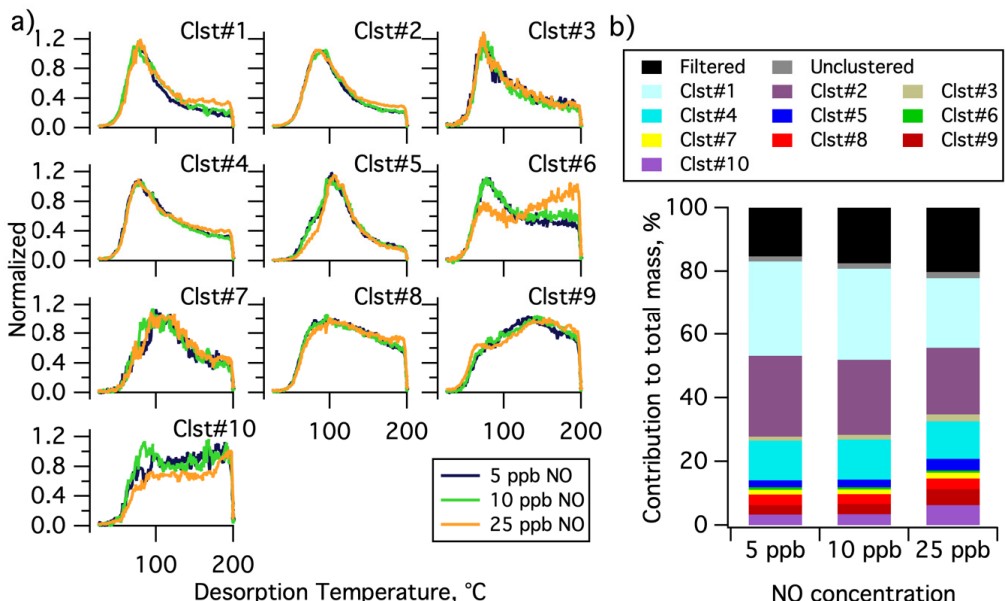

1119

**Figure 9.** Single clustering results for α-pinene + OH SOA as a function of NO concentration. (a) Comparison of the normalized, weighted average thermograms of the ten clusters for the 5 ppb NO (navy), 10 ppb NO (green) and 25 ppb NO (orange) experiments. (b) Contribution of each cluster to the total mass, including the contribution from filtered out ions (black) and unclustered ions (gray). The total mass is calculated independently for each experiment.


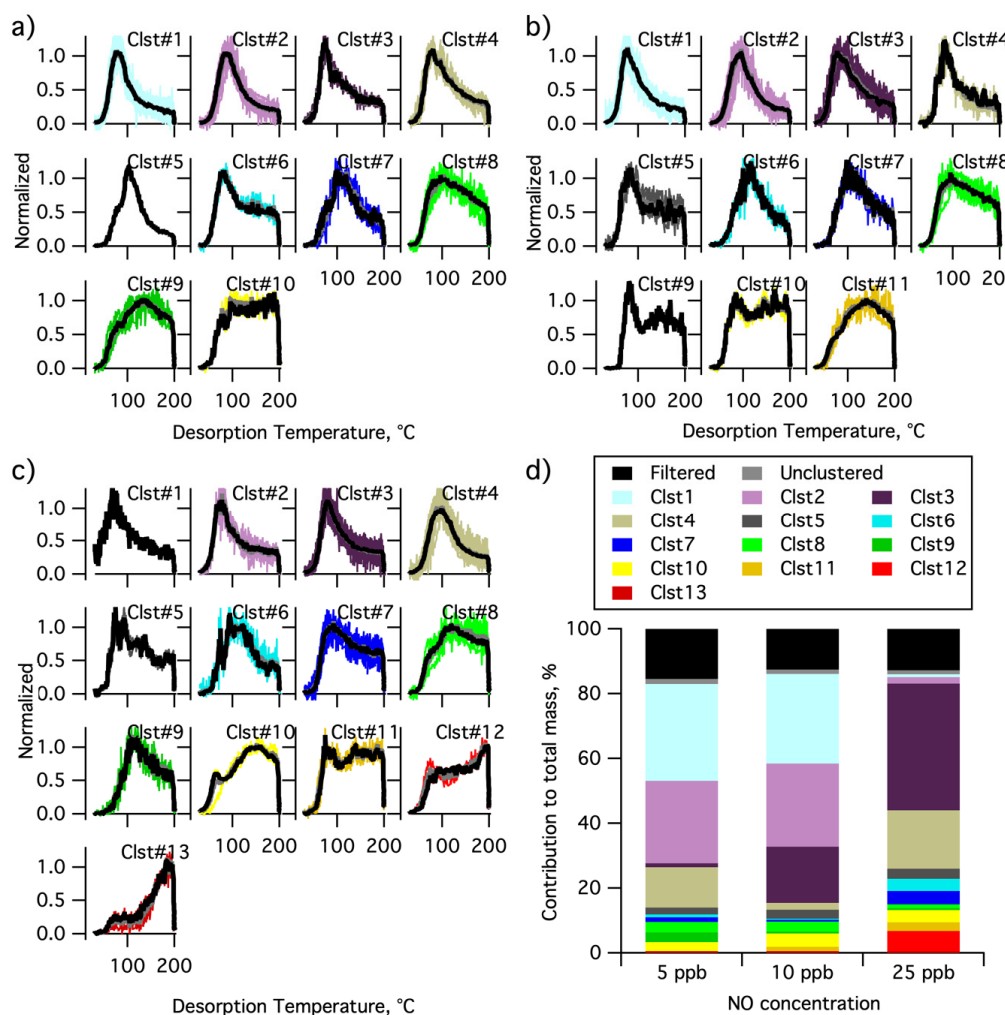

**Figure 10.** Multiple clustering results for α-pinene + OH SOA as a function of NO concentration. Clustering results are separately shown for the (a) 5 ppb NO, (b) 10 ppb NO, and (c) 25 ppb NO experiments. Each panel includes unweighted average thermograms (grey lines), mass-weighted average thermograms (black lines) and individual cluster members (colored lines). (d) Contribution of each cluster to the total mass for each experiment. The mass contribution of filtered-out ions (black bar) and unclustered ions (gray bar) are also shown.





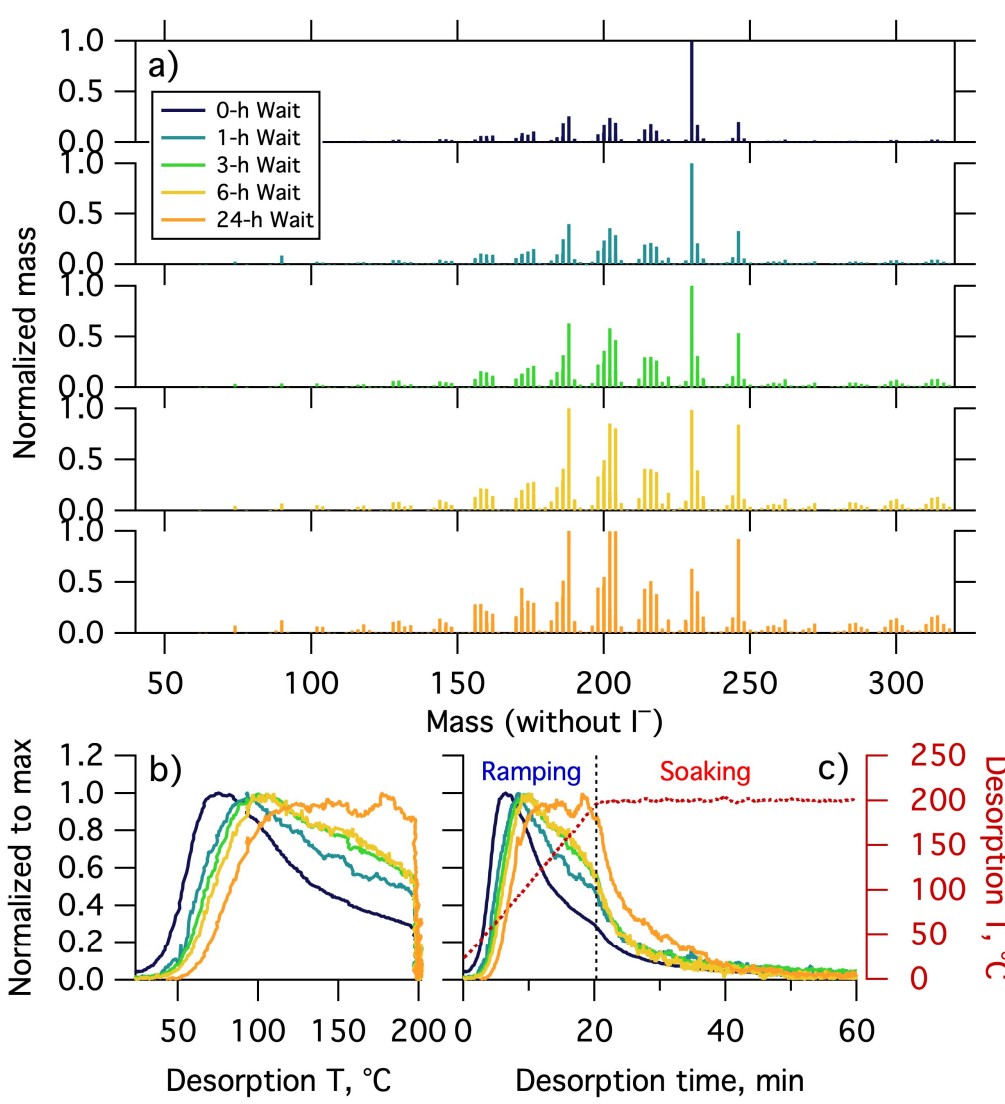


**Figure 11.** (a) Normalized mass spectra of α-pinene + O₃ SOA measured after different extents of isothermal evaporation at room temperature. The mass excludes iodine. The normalized thermograms of bulk SOA versus (b) temperature and (c) time, with the desorption temperature shown as a red dashed line. The vertical black dashed line in (c) delineates between ramping and soaking. The mass spectrum or thermogram colors indicate the isothermal evaporation time (see legend), with darker colors indicating shorter times.

1142

**Figure 12.** Single clustering results for α-pinene + O₃ SOA for different isothermal evaporation times. (a) Comparison of the normalized, weighted-average thermograms of the 12 clusters of 0-h wait (navy), 1-h wait (blue), 3-h wait (green), 6-h wait (yellow) and 24-h wait (orange) experiments. Note that the absolute signals of all of the clusters decrease with evaporation, but to varying extents (**Figure S6**).



**Figure 13.** Multiple clustering results for $\alpha$-pinene + O$_3$ SOA as a function of isothermal evaporation time. (a) Contribution of each cluster to the total mass for each experiment, along with the contributions of filtered-out ions (black bar) and unclustered ions (gray bar). The number of clusters obtained generally decreases with isothermal evaporation time. (b-f) The unweighted average (gray) and mass-weighted average (black) thermograms, along with the thermograms of individual members of clusters for the (b) 0-h, (c) 1-h, (d) 3-h, (e) 6-h, and (f) 24-h wait experiments. The cluster colors are consistent between panels.



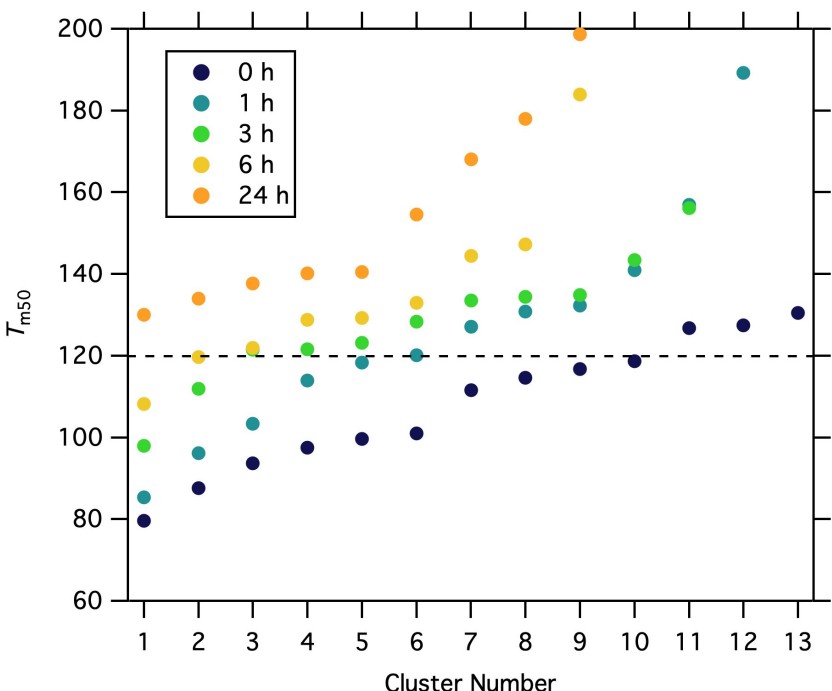


**Figure 14**. The $T_{m50}$ values of the cluster-specific thermograms from multiple clustering for the five
isothermal evaporation experiments.

