# Peer review of "A robust clustering algorithm for analysis of composition-dependent"

_Atmospheric Chemistry and Physics, 2019_

## Referee Comment (RC1) · Anonymous Referee #1 · 7 Oct 2019

The authors present a well structured detailed report of method they are proposing to analyse thermograms collected using FIGAERO-CIMS data. Although the manuscript focusses on cluster analysis, it clear that a considerable amount of work has gone into collecting the data, and developing and trialling the method (Noise-Sorted Scanning Clustering). It is difficult to find fault in the work. Their introduction gives a good panorama of the cluster analysis and air quality data vista. They select the various suitable cluster analysis methods and make comparisons using their data before justifying their choice of NSSC. The data flow is well described and supported by illustrations and them exemplified by application to laboratory generated SOA. It will be interesting to see how this method deals with ambient data.

[Figure]

The following points are simply minor considerations on how to improve the presentation taking into consideration this is a paper on a new method of clustering and as a reader, I am asking if I could reproduce this method for a different application or in a different code.

1. I appreciate the descriptions given of the methods and especially figure 2 and I am asking myself if more detailed mathematics be included to describe the method?

2. I can see that there is a lot of information conveyed in figures 5, 7, 9, 10 and especially 13 and I am asking myself if they can be enhanced to better convey their message?
* * *

---

## Referee Comment (RC2) · Anonymous Referee #2 · 7 Oct 2019

Note: This review is focused on the clustering aspects of the paper only. I make no comment on whether the generated clusters constitute a useful and valid contribution to the analysis of composition‐dependent 2 organic aerosol thermal desorption measurements as this is not in my area of expertise. I leave that for others to comment on.

From the title and abstract I was expecting to read about a new clustering algorithm. However, this paper presents a data pre-processing step before using DBScan, followed by an application-specific post-clustering process.

**Summary** The section on the clustering process should be re-written. This is not a new

clustering algorithm, but a method of pre- and post-processing data to provide useful analysis. I do not deny the usefulness of the analysis, its methodology or results. There are also points below that should be clarified.

1. DBScan is compared to other clustering algorithms, however these alternatives are significantly different in their intended use, i.e. they generate hype-ellipsoidal clusters, whereas DBScan is specifically targeted at arbitrarily shaped clusters. It would be better to compare DBScan with similar algorithms. For the purposes of this paper, I am not sure that comparison to other algorithms is required.

2. Line 111 states 'a novel variant of DBScan'. The clustering algorithm used appears unchanged from DBScan, but rather a pre-processing technique of ordering the data is utilised before clustering.

3. Line 132 'absolute magnitude' is moot, magnitude has no direction.

4. The paragraph from lines 134-144 describes a process based on a number of factors which are not justified. This leaves the questions: why 100 points; why +/- 50 points; and why is an anomaly only outside of $-3\sigma$ and not outside of $+3\sigma$ also?

5. Similarly, lines 171-182 use a smoothing over 35 points. It is not clear how this smoothing is carried out, e.g. a mean of 35 points? Is it +17/-18, or -35, or +35 points that are smoothed? If the thermograms have peaks around 40 points wide are we seeing the mean of the value in the peak? If so how does this correlate with 'retaining the peak shape'? I think this should be clarified.

6. Similarly, line 190 recommends a weighting of 4:1. How has this value been arrived at?

7. Lines 216-218 discuss the removal of noise. DBScan is considered deterministic for core points and noise. Noisy data would normally be identified and be members of clusters $< minpts$. Is there a danger that data identified with 'high levels of noise' is excluded when, despite the noise, it is similar enough to be included in a cluster?

8. Lines 226-235 appear to form the work being considered as 'a novel variant of DBScan'. This describes DBScan with no alterations, except to force the order of data to consider 'seed thermograms' first. This is a pre-processing stage of the data, rather than a novel clustering algorithm. DBScan is deterministic if data order is preserved. If data order is not preserved then it is deterministic for core and noise, with only border points varying in some cases. I'd be interested to see how the border points vary to justify forcing the data order. Are the results generated by forcing the data order better simply because they consistent with each run, whereas random initialisation is not? If so, is it possible to identify which thermograms change cluster and consider why?

9. Lines 240-255 describe the DBScan algorithm. I am unclear how this varies from standard DBScan. Cycling through those thermograms identified as 'seeds' could equally be done by ordering the data by order of noise, then using the data, in order, to run standard DBScan.

10. Line 240 refers to Figure 2. This is not a suitable method for presenting an algorithm and a formal pseudo-code should be used. This may help clear up any confusion over the similarities or differences between DBScan and the method proposed.

11. Lines 256 – 269 describe a 'second round' of clustering. This generates new data for each cluster in the form of a 'signal weighted average', presumably of the cluster members. A thermogram that is within $\epsilon$ of the average, but not already clustered suggests that it is a border point, but below the $minpts$ threshold for

inclusion? (I am unclear how a thermogram can be within $\epsilon$ of the average, but not within $\epsilon$ of $minpts$ of other cluster members?) This part appears to be a novel 'second stage', however I would not consider this to be a clustering algorithm in itself, but rather a post-processing step to tidy up 'stragglers', which is application specific.

12. Section 2.2.3 Describes a process for selecting an optimal $\epsilon$ value. The selection of $\epsilon$ is based on fuzzy terms such as 'small' and 'near the maximum'. Figure 2 shows a clear value of $\epsilon$ in this case, is this the same for all analyses?

13. I am also unclear from section 2.2.3 whether this selection of optimal $\epsilon$ is generic to all future datasets of this type, or whether this optimal selection process is required for each new set of data?

---

## Author Comment (AC1) · 19 Nov 2019

We thank the reviewers for the thoughtful comments. We address each comment individually below, with the reviewers' initial comment in **black** and our responses in **blue**. A track changes version showing all changes made to the manuscript is appended at the end.

**Response to Reviewer #1**

The authors present a well structured detailed report of method they are proposing to analyse thermograms collected using FIGAERO-CIMS data. Although the manuscript focusses on cluster analysis, it clear that a considerable amount of work has gone into collecting the data, and developing and trialling the method (Noise-Sorted Scanning Clustering). It is difficult to find fault in the work. Their introduction gives a good panorama of the cluster analysis and air quality data vista. They select the various suitable cluster analysis methods and make comparisons using their data before justifying their choice of NSSC. The data flow is well described and supported by illustrations and them exemplified by application to laboratory generated SOA. It will be interesting to see how this method deals with ambient data. The following points are simply minor considerations on how to improve the presentation taking into consideration this is a paper on a new method of clustering and as a reader, I am asking if I could reproduce this method for a different application or in a different code.

We first thank the reviewer for the positive assessment. We think the NSSC method is fairly easy to transfer to a different code. The application of NSSC can be potentially expanded to any composition-resolved data sets, such as diurnal changes of different compounds measured in ambient air, temporal changes of different generations of species in a smog chamber, and composition-dependent size distributions. All of the above data sets share a common property that the noise of the curve/spectrum is related to the composition. We have expanded the discussion of the application of NSSC in section 5.

"This paper focuses only on the description of the clustering algorithm and its potential as a tool to characterize the *thermal* properties of organic aerosol in further detail. *The application of NSSC can be potentially expanded to any other composition-resolved data sets, such as diurnal changes of different compounds measured in ambient air, temporal changes of different generations of species in a smog chamber, and composition-dependent size distributions. All of the above data sets share a common property that the noise of the curve/spectrum is related to the composition. Therefore, NSSC would facilitate the analysis by taking noise into consideration.*"

To the reviewers point that this will be interesting to see how the method deals with ambient data, we agree and are actively pursuing this idea.

1. I appreciate the descriptions given of the methods and especially figure 2 and I am asking myself if more detailed mathematics be included to describe the method?

The reviewer raises an important point. Figure 2 serves to provide an overview of the flow of NSSC method to help readers understand the algorithm. Therefore, we made figure 2 a more generic description. We described all of the detailed mathematics in the text only because most of these parameters are ultimately data-specific and user-defined.

2. I can see that there is a lot of information conveyed in figures 5, 7, 9, 10 and especially 13 and I am asking myself if they can be enhanced to better convey their message?

The reviewer raises an important point. There are indeed a lot of information in each of the figure the reviewer mentions. For figure 5 and 7, we think they are the best way to provide clustering results for simple single-precursor SOA systems at the moment. We have tried to plot all the average cluster thermograms in one figure. However, due to the existence of many overlaps, it makes the comparison between different clusters more difficult. We have also tried to use pie chart instead of bars to show the percentage contribution of clusters. We think they are equally efficient in conveying information and the bar chart is more convenient when different systems are compared in one figure. For figure 10 and 13, they show the clustering results of a set of experiments using the multiple clustering approach. It is necessary to show all the averaged thermograms of all the experiments in order to find which of the thermograms is common in different experiments while which of the thermogram is unique in only one experiment. As for the bar charts, it would be clearer to show the percentage contribution of grouped clusters based on $T_{m50}$ as is described in section 4.4.2. However, we choose to leave the original, detailed information in the figure to both give an example of the detailed clustering results of NSSC and let the readers explore further interpretations of the clustering results. Therefore, we have not made modifications to these figures. We note that we have provided the information from these figures in an accessible, downloadable format with the associated dataset so that readers can explore the data further.

**Response to Reviewer #2**

Note: This review is focused on the clustering aspects of the paper only. I make no comment on whether the generated clusters constitute a useful and valid contribution to the analysis of composition-dependent organic aerosol thermal desorption measurements as this is not in my area of expertise. I leave that for others to comment on.
From the title and abstract I was expecting to read about a new clustering algorithm. However, this paper presents a data pre-processing step before using DBScan, followed by an application-specific post-clustering process.

**Summary** The section on the clustering process should be re-written. This is not a new clustering algorithm, but a method of pre- and post-processing data to provide useful analysis. I do not deny the usefulness of the analysis, its methodology or results. There are also points below that should be clarified.

We first thank the reviewer for the thoughtful comments on the clustering algorithm. The reviewer's comments have helped us to clarify aspects of the manuscript, making clearer the unique aspects of this work.

We agree that the NSSC stems from DBScan, and noted as much in the text when we state that the NSSC is "a novel variant of the DBSCAN algorithm." However, we contend that the NSSC differs from DBScan in important ways besides the process of seeds sorting and second-round clustering the reviewer mentions, making it sufficiently "new." For example, the way NSSC and DBSCAN define a cluster differ. Details will be described in the following responses. (Perhaps semantics, but we would also contend that the "preprocessing" and "post-clustering process" constitute part of the overall algorithm. A definition of algorithm is "a process or set of rules to be followed in calculations or other problem-solving operations, especially by a computer.") At the current stage, NSSC is designed specifically for FIGAERO-CIMS

thermograms, but has the potential to be applied to other composition-specific data sets. The values of many parameters and factors used in the data processing are empirically derived. Ultimately, they can be adjusted by users based on their specific applications. All this said, we have made revisions to the manuscript to more clearly indicate the link to the DBScan algorithm, including in the abstract.

1. DBScan is compared to other clustering algorithms, however these alternatives are significantly different in their intended use, i.e. they generate hype-ellipsoidal clusters, whereas DBScan is specifically targeted at arbitrarily shaped clusters. It would be better to compare DBScan with similar algorithms. For the purposes of this paper, I am not sure that comparison to other algorithms is required.

The reviewer raises an important point about the comparison of different clustering algorithms. As the reviewer noted, DBScan and other algorithms have different intended use. For the FIGAERO-CIMS thermograms, however, it is difficult to define the data as hyper-ellipsoidal clusters or arbitrarily shaped clusters. Therefore, we tried DBScan, k-means, k-medoids and mean-shift, as they are well-known and most commonly used clustering algorithms. A brief description and comparison of these methods are presented in section 2.3 to give a context of why we chose NSSC. We believe this comparison provides value, especially as the only other attempt at clustering FIGAERO data that we are aware of used k-means.

2. Line 111 states 'a novel variant of DBScan'. The clustering algorithm used appears unchanged from DBScan, but rather a pre-processing technique of ordering the data is utilised before clustering.

The reviewer points out that the seed-sorting process of NSSC should not be considered as a variant of DBScan. We understand that this distinction is important to make, although contend that the "algorithm" encompasses the entire set of standardized procedures used, including pre-processing. By adding the pre-processing and post-processing steps, the NSSC is definitionally a "variant." (Variant, *noun*, "a form or version of something that differs in some respect from other forms of the same thing or from a standard.") We also note that there is an additional aspect that makes NSSC different from DBScan. To the best of our knowledge, in a cluster defined by DBScan, there are a core point, directly reachable points and reachable points. Directly reachable points are within the criterial distance $\varepsilon$ of the core point, while reachable points are within the distance $\varepsilon$ of directly reachable points or other reachable points. The inclusion of reachable points is the key reason why DBScan can find arbitrarily shaped clusters. However, NSSC only considers the core point (seed) and the directly reachable points (neighbors) as a cluster in the first step. There is a second step of clustering where the seed is redefined based on the cluster average thus far and new directly reachable points added, expanding the number of members that are included in a given cluster. However, the number of new members added in this second step tends to be small. In some ways, NSSC is more similar to for example k-means in a way it generates hype-ellipsoidal clusters. We have clarified the difference between NSSC and DBScan in section 2.3.

"Noisy members tend to naturally be excluded from any clusters. *NSSC is a variant of DBSCAN. It does, however, differ from the standard DBSCAN algorithm because NSSC only searches for neighbors of the seed, while DBSCAN also searches for neighbors of the neighbors. As such, the sorting of seeds by noise levels is a key aspect of the NSSC algorithm which we have found provides for more robust clustering results.*"

3. Line 132 'absolute magnitude' is moot, magnitude has no direction.

We have deleted "absolute".

4. The paragraph from lines 134-144 describes a process based on a number of factors which are not justified. This leaves the questions: why 100 points; why +/- 50 points; and why is an anomaly only outside of -3∗σ and not outside of +3σ also?

The reviewer raises an important point about justification of several values used in the pre-processing analysis. These values are derived based on consideration of 10 different smog chamber experiments with different chemical systems but similar FIGAERO-CIMS operation. The number of points we determined should be used for noise determination (100 points) derives from inspection of the thermograms from the different experiments and our finding that, as stated, during this period the "signals are usually relatively constant." Use of more points leads to undesirable incorporation of times when the signals are still declining during the soaking period, increasing the standard deviation. Use of fewer points leads to larger overall noise levels. We have added statements to this effect to the manuscript. The number of points used to determine the minimum signal (+/-50) was determined based on the temperature ramping speed and a desire to identify as "noisy" only those thermograms that exhibited large negative deviations. We established that use of many fewer points led to an over-sensitivity to small fluctuations. Use of a greater number of points led to excessive smoothing. Additionally, the selection of +/- 50 points for calculating the minimum provides consistency with the number of points averaged for determining the noise. An anomaly refers to only outside of -3σ because the values in a thermogram are all background corrected and expected to be positive. So, $A_{min} < -3\sigma$ indicates that the minimum is at least three standard deviations below zero. We have added a discussion of the values of factors used in this paper at the beginning of section 2.1.1 to clarify.

"Estimate a reference noise level ($\sigma_{ref}$) for each thermogram as the standard deviation of the last 100 points (corresponding to 500 seconds) of the thermogram at the end of the constant-temperature soaking period, during which the signals are usually relatively constant. *Use of more points incorporates times when the signals were still decreasing, while use of fewer points provides a less robust estimate of the noise level.* (ii) Find the minimum in the thermogram and calculate the average of this and the 50 points (corresponding to 250 seconds, *or 100 points*) before and after the minimum, $A_{min}$. *This provides for consistency with the determination of* $\sigma_{ref}$ (iii) Identify thermograms for which $A_{min} < -3*|\sigma_{ref}|$ as anomalous and exclude these associated ions from further analysis. In other words, when a thermogram has a valley with averaged negative values exceeding the magnitude of three times of the reference noise level, then it is considered anomalous. *The specific criteria specified above were determined based on consideration of thermograms from 10 distinct SOA experiments. While these criteria should be robustly applicable to other FIGAERO-CIMS datasets, they can be adjusted depending on the specific application, data quality, and needs.*"

5. Similarly, lines 171-182 use a smoothing over 35 points. It is not clear how this smoothing is carried out, e.g. a mean of 35 points? Is it +17/-18, or -35, or +35 points that are smoothed? If the thermograms have peaks around 40 points wide are we seeing the mean of the value in the peak? If so how does this correlate with 'retaining the peak shape'? I think this should be clarified.

A boxcar smoothing function is used, where the average is for the central point and the (N-1)/2 points before and after the point, where N is the number of points in the boxcar, here 35; this applies to all points that are at least 1+(N-1)/2 points from the edges (the first and last point). We now state that a boxcar smoothing function is used. Points that are closer to the edges than 1+(N-1)/2 points are excluded. Thus, the smoothed thermogram has N-1 fewer points than the unsmoothed, here a total of 34 out of 800. Since peaks are almost never observed at the start or end of the thermogram this exclusion of points does not impact the determination of the smoothed maximum for each thermogram. As to "retaining the peak shape," such smoothing only occasionally leads to lowering of the especially sharp peaks, however, will retain the location and shape of most peaks. We also show an example smoothed thermogram, compared to the original thermogram, below. This example is a reasonable sharp thermogram, to illustrate that the shape is nominally preserved, although the maximum is certainly reduced relative to the single-point maximum. We have not included such a figure in the manuscript, although have expanded the description of the smoothing in section 2.1.2.

"Normalization is achieved by dividing each point of the original thermogram by the thermogram maximum, *where the maximum is determined* after smoothing using a 35-point *boxcar* moving average *with the end points excluded from the smoothed thermogram.*"

[Figure]

6. Similarly, line 190 recommends a weighting of 4:1. How has this value been arrived at?

The value is empirical based on the information two different heating stages carry. We have also tried 1:1, 2:1 and 10:1 for the example data sets, yet 4:1 leads to the best clustering results. This value is ultimately user-defined depending on what kind of information they are trying to extract from the thermograms. We have expanded the discussion regarding this weighting factor.

"we recommend down-weighting the soaking period such that the ramping and soaking periods ultimately carry approximately 4:1 weight in the calculation of the *ED. We have tested weighting of 1:1, 2:1 and 10:1. Weighting of 4:1 provides for the most robust clustering results for the example datasets.*"

7. Lines 216-218 discuss the removal of noise. DBScan is considered deterministic for core points and noise. Noisy data would normally be identified and be members of clusters < minpts. Is there a danger that data identified with 'high levels of noise' is excluded when, despite the noise, it is similar enough to be included in a cluster?

The reviewer raises an important point about the treatment of noisy thermograms in the data processing stage. As we described in section 2.1.3, the noise level is inversely related to the signal level of an ion. Therefore, the excluded noisy data are usually unimportant and makes up only a small fraction of total signal. For all the example data sets, we have not discovered any excluded noisy thermogram carrying significant mass that also has a shape clearly similar to the clustered thermograms. We note that one reason for excluding the noisy thermograms is that this defines a criteria that helps guide inspection of the single-ion clusters (i.e., those having fewer than minpts). Similar to DBSCAN, the NSSC also has the ability to identify noisy data or outliers due to the insert of seed-sorting process. In most cases, NSSC with and without a pre-removal of noisy data gives identical clustering results. The user can choose to skip the treatment of noisy data, if desired. We have added the following as additional context regarding removal of noisy thermograms.

"This is especially the case for algorithms such as k-means and partitioning around medoids, which assign all the members to a cluster. *Clustering methods that do not require assignment of all members, such as DBSCAN or our NSSC, are generally less sensitive to the influence of overly noisy members. However, we have found that the explicit exclusion of noisy thermograms up front serves to provide for more robust behavior and also removes the need to consider each noisy thermogram as a possible single-member cluster."*

8. Lines 226-235 appear to form the work being considered as 'a novel variant of DBScan'. This describes DBScan with no alterations, except to force the order of data to consider 'seed thermograms' first. This is a pre-processing stage of the data, rather than a novel clustering algorithm. DBScan is deterministic if data order is preserved. If data order is not preserved then it is deterministic for core and noise, with only border points varying in some cases. I'd be interested to see how the border points vary to justify forcing the data order. Are the results generated by forcing the data order better simply because they [are] consistent with each run, whereas random initialisation is not? If so, is it possible to identify which thermograms change cluster and consider why?

As we discussed in the response to comment #2, the clustering algorithm of NSSC also differs from that of DBScan in the way they determine neighbors with a seed. We have expanded the discussion of the differences of NSSC and DBScan in section 2.3 and added the following statement to the description of the NSSC regarding the treatment of the seed: "*The seed does not evolve as neighbors are added to the cluster during this step.*" For the same dataset, NSSC generally results in more clusters. In NSSC there will not be border points, unlike DBScan, because there is only one time of scanning for each seed.

9. Lines 240-255 describe the DBScan algorithm. I am unclear how this varies from standard DBScan. Cycling through those thermograms identified as 'seeds' could equally be done by ordering the data by order of noise, then using the data, in order, to run standard DBScan.

As we discussed in the response to comment #2, the clustering algorithm of NSSC also differs from that of DBScan in the way they determine neighbors with a seed, besides the inclusion of seed-sorting and second-round of clustering processes.

10. Line 240 refers to Figure 2. This is not a suitable method for presenting an algorithm and a formal pseudo-code should be used. This may help clear up any confusion over the similarities or differences between DBScan and the method proposed.

It is our understanding that flowcharts are generally accepted as a way to present algorithms. Flowcharts for algorithm presentation have reasonably standardized symbols and structure, which we have endeavored to follow. When deciding between whether to include a flowchart or pseudocode, we considered whom we thought most likely to read and use this work and concluded the target audience is atmospheric chemists and other FIGAERO-CIMS users. As atmospheric chemists are generally familiar with flowcharts, but typically have less familiarity with pseudocode, we decided that presentation of our algorithm as a flowchart was preferable. We understand the reviewers point that clear distinction between DBScan and NSSC is necessary, and have added to the text to help in this regard, as discussed above. We note finally that the complete code of NSSC is also available at GitHub (https://github.com/chriscappa/NSSC).

11. Lines 256 – 269 describe a 'second round' of clustering. This generates new data for each cluster in the form of a 'signal weighted average', presumably of the cluster members. A thermogram that is within $\varepsilon$ of the average, but not already clustered suggests that it is a border point, but below the *minpts* threshold for inclusion? (I am unclear how a thermogram can be within $\varepsilon$ of the average, but not within $\varepsilon$ of *minpts* of other cluster members?) This part appears to be a novel 'second stage', however I would not consider this to be a clustering algorithm in itself, but rather a post-processing step to tidy up 'stragglers', which is application specific.

As we discussed in the response to comment #2, the clustering algorithm of NSSC is different from that of DBScan in the way they determine neighbors with a seed. Therefore, there is no concept of "border points" in NSSC. Since the average thermogram of a cluster can be slightly different from seed thermogram, this second round of clustering is necessary to tidy up stragglers. The algorithm of the second round is more similar to how mean-shift method adjust the existing clusters. Therefore, we think this should be considered to be part of the algorithm.

12. Section 2.2.3 Describes a process for selecting an optimal $\varepsilon$ value. The selection of $\varepsilon$ is based on fuzzy terms such as 'small' and 'near the maximum'. Figure  4 shows a clear value of $\varepsilon$ in this case, is this the same for all analyses?

The reviewer raises an important point on the determination of optimal $\varepsilon$. In order to find the optimal $\varepsilon$, NSSC has to be run on the dataset for multiple times for a range of $\varepsilon$. We have shown in figure 4 as an example of how to determine the optimal $\varepsilon$. There are four parameters exhibiting different behaviors as a function of $\varepsilon$. At the current stage, we recommend users to determine optimal $\varepsilon$ by visually comparing the values of these four parameters at different $\varepsilon$ with the help of a figure such as figure 4. In the future, it would be ideal to find a way to determine the optimal $\varepsilon$ automatically. We note that in leaving the decision about the optimal $\varepsilon$ "fuzzy," the approach here shares some similarity with other clustering algorithms for which, for example, the number of clusters or the minimum epsilon must be specified. It is our understanding that various approaches, such as "elbow plots" have been used to determine optimal parameters, but quite often these have an element of "fuzzy"-ness to them, as here. As to the reviewer's question about whether there is a clear epsilon for all cases, we provided the determination plots for every experiment considered in the supplemental material. Fig. 4 is one example.

13. I am also unclear from section 2.2.3 whether this selection of optimal $\varepsilon$ is generic to all future datasets of this type, or whether this optimal selection process is required for each new set of data?

The selection of $\varepsilon$ is specific to each experiment and dataset. However, the guidance to determine an optimal value is generic. The determination plots, similar to Fig. 4, for each dataset are provided in the Supplemental Material.

[revised manuscript text omitted]

---

## Author Response (AR2)

We thank the editor for the thoughtful comments. We address each comment individually below, with the editor's initial comment in **black** and our responses in **blue**.

**Response to the editor**

In the abstract, I would like you to endeavour to be very explicit in distinguishing clusters of mass
spectra versus clusters of molecules forming aerosol. I leave it to you to find the best language,
but suggest 'clusters of spectra' and 'aerosol particles' are used throughout the abstract, for
clarity. I realise this may seem heavy-handed, but I'm concerned especially for inexperienced
researchers whose first language may not be English.
We have added "mass spectral" to describe clusters.
Line 86, please delete "a new clustering method" and change 'a novel extension' to 'an extension'.
Line 116: please delete 'novel'
Line 312: please delete 'if any' (I think the meaning is already captured in the parentheses
immediately beforehand).
We have deleted the words accordingly.
At the end of section 2.2, I would welcome a paragraph summarising the truly automated process
and the part requiring human input (and when this input is for every experiment or just for every
new kind of experiment). Please could you think about how you might convey an estimate of the
time saved? Or is it not so much time saved as patterns found objectively? If that is the case,
please say so.
We have added a paragraph as section 2.2.4 to summarize the algorithm.
Line 407ff. I think more is needed here about why it is better to look only at neighbours of the
seed, rather than neighbours of the neighbours. Can elaborate your argument that NSSC is a
development of DBScan and make sure it does not appear to be a simplification requiring quite
a lot of data pre- and post-processing?
We have added several sentences in section 2.3 to address the concern.

[revised manuscript text omitted]

* Experiment #1 is a case study used to test the performances of different clustering algorithms
[#] Conc. of precursors are the concentrations expected in the chamber with the absence of any chemistry
[##] For OH, conc. refers to concentration of H$_2$O$_2$ injected into the chamber; for O$_3$, conc. refers to steady-state concentration of O$_3$ in the chamber during SOA formation
[#*] Seed particles are size-selected in all the experiments
[#$] NO concentration refers to the targeted NO concentration when NO is injected into the chamber. The actual steady-state concentration of NO is lower than targeted. "-" indicates that no external NO is added to the chamber
[#&] $M_p$ is the estimated mass concentration of particles including SOA and seeds measured by SMPS when the chamber is at steady-state, except for experiment 4 where $M_p$ is the mass concentration of SOA only
[$] Normal operation mode means the desorption process starts immediately after collection period. X h wait means that particles are isothermally diluted for X hours before the desorption process is initiated
[&] AS = ammonium sulfate
[&&] PS = potassium sulfate

**Table 2.** Comparison of different clustering algorithms

| Clustering Algorithms | k-means | k-medoids | Mean-shift | DBSCAN | FPClustering | NSSC |
|---|---|---|---|---|---|---|
| Assign all the members? | Yes | Yes | No | No | Yes | No |
| Identify single-member clusters? | No | No | Yes | No | No | Yes |
| Robust solution? | No | No | No | Yes | No | Yes |
| Controlled distance from the center of clusters? | No | No | Yes | No | No | Yes |
| Influence of noise? | large | large | small | small | large | Small |
| Key preset parameters | $N_c$ | $N_c$ | $\varepsilon, N_{min}$ | $\varepsilon$ | Initial seed | $\varepsilon, N_{min}$ |
| Software used in this study | Igor | R | Python | Igor | Igor | Igor |

**Table 3**. Parameters and thresholds used for the data processing and noise-sorted scanning clustering for
all the example experiments.

[revised manuscript text omitted]